# Spatiotemporal dynamics of macrophage heterogeneity and a potential function of Trem2hi macrophages in infarcted hearts

Seung-Hyun Jung [1,2,9] ✉, Byung-Hee Hwang[3,4,9], Sun Shin [5,9], Eun-Hye Park[3], Sin-Hee Park [3], Chan Woo Kim [3], Eunmin Kim[3], Eunho Choo[3,4], Ik Jun Choi[6], Filip K. Swirski [7], Kiyuk Chang [3,4] ✉ & Yeun-Jun Chung [2,5,8] ✉

Heart failure (HF) is a frequent consequence of myocardial infarction (MI). Identification of the precise, time-dependent composition of inflammatory cells may provide clues for the establishment of new biomarkers and therapeutic approaches targeting post-MI HF. Here, we investigate the spatiotemporal dynamics of MI-associated immune cells in a mouse model of MI using spatial transcriptomics and single-cell RNA-sequencing (scRNA-seq). We identify twelve major immune cell populations; their proportions dynamically change after MI. Macrophages are the most abundant population at all-time points (>60%), except for day 1 post-MI. Trajectory inference analysis shows upregulation of *Trem2* expression in macrophages during the late phase post-MI. In vivo injection of soluble Trem2 leads to significant functional and structural improvements in infarcted hearts. Our data contribute to a better understanding of MI-driven immune responses and further investigation to determine the regulatory factors of the Trem2 signaling pathway will aid the development of novel therapeutic strategies for post-MI HF.

With the technical advances and widespread adoption of cardiac revascularization strategies and guideline-directed medical therapeutic interventions, the age-standardized death rates and heart failure hospitalization rates after acute myocardial infarction (AMI) have been gradually declining[1,2]. However, the rate of recurrent events, including heart failure-associated hospitalization and death after myocardial infarction (MI) are still very high. Heart failure is a common condition among individuals who survived an MI attack, mainly due to adverse left ventricular remodeling[3,4].

The ischemic injury following AMI induces the mobilization and recruitment of a diverse repertoire of innate and adaptive immune cells to the infarcted heart[5]. Of these, macrophages are critical for the clearance of infarcted tissues as well as for wound repair and remodeling processes[6]. Macrophages are broadly divided into M1 and M2 subtypes according to their in vitro construction; M1 macrophages express high levels of pro-inflammatory cytokines and promote a pro-inflammatory milieu, whereas M2 macrophages release anti-inflammatory cytokines and promote angiogenesis and wound healing. The balance between M1 and M2 macrophages during immune

[1]Department of Biochemistry, College of Medicine, The Catholic University of Korea, Seoul 06591, Republic of Korea. [2]Department of Biomedicine & Health Sciences, College of Medicine, The Catholic University of Korea, Seoul 06591, Republic of Korea. [3]Cardiovascular Research Institute for Intractable Disease, College of Medicine, The Catholic University of Korea, Seoul 06591, Republic of Korea. [4]Division of Cardiology, Seoul St. Mary's Hospital, College of Medicine, The Catholic University of Korea, Seoul 06591, Republic of Korea. [5]Departments of Microbiology, IRCGP, Cancer Evolution Research Center, College of Medicine, The Catholic University of Korea, Seoul 06591, Republic of Korea. [6]Division of Cardiology, Incheon St. Mary's Hospital, College of Medicine, The Catholic University of Korea, Seoul 06591, Republic of Korea. [7]Cardiovascular Research Institute, Icahn School of Medicine at Mount Sinai, New York, NY, USA. [8]Precision Medicine Research Center, College of Medicine, The Catholic University of Korea, Seoul 06591, Republic of Korea. [9]These authors contributed equally: Seung-Hyun Jung, Byung-Hee Hwang, Sun Shin. ✉e-mail: hyun@catholic.ac.kr; kiyuk@catholic.ac.kr; yejun@catholic.ac.kr

responses was defined as essential for effective healing and remodeling processes[6]. However, this dichotomous classification was obsolete, as in vivo environment of macrophages are more complex[7]. Therefore, a more precise evaluation is needed for accurate characterization of the dynamics of macrophage heterogeneity during the acute period of MI. Single-cell RNA-sequencing (scRNA-seq) allows the investigation of the transcriptional states of heterogeneous cell populations at a high resolution and has already been applied to non-myocyte cardiac cell populations of healthy mice[8] and immune cell populations of mice hearts after MI[9–12].

In this study, we perform spatial and scRNA-seq analyses of cardiac immune cells in a mouse MI model to investigate the spatio-temporal dynamics of MI-associated immune cells. We identify a macrophage subset, Trem2[hi] macrophages, with anti-inflammatory characteristics, specifically dominant in the late-stage infarcted heart. Moreover, in vivo injection of soluble Trem2 leads to significant functional and structural improvements in infarcted hearts. Overall, our data may enable the identification of biomarkers and the development of therapeutic strategies for MI.

## Results

### Immune cell dynamics after MI

To obtain a comprehensive landscape of immune cells after ischemic injury of the heart, we performed scRNA-seq on flow cytometry-sorted CD45[+] leukocytes isolated from infarct and peri-infarct areas 1, 3, 5, and 7 days after the induction of MI in mice (Fig. 1a). CD45[+] cells in the steady-state (before MI induction) were also used as a control for scRNA-seq. A total of 33,977 cells captured after quality control filtering (steady-state: 6581 cells; day 1 post-MI [day 1]: 7363 cells; day 3 post-MI [day 3]: 4446 cells; day 5 post-MI [day 5]: 7271 cells; day 7 post-MI [day 7]: 8316 cells) were integrated, clustered, and visualized in uniform manifold approximation and projection (UMAP) plots using the Seurat R package[13].

To annotate cell clusters, the SingleR package[14] was used to assess the expression of well-known cell-type-specific markers; a total of 12 broad cell clusters were defined within CD45[+] cells (Fig. 1b). The most abundant cell population was macrophages (64.2% of total CD45[+] cells); they highly expressed prototypical macrophage genes such as *Csf1r*, *Cd68*, and *Adgre1* (Fig. 1c and Supplementary Data 1). Non-macrophage cell populations included three clusters of dendritic cells (DC: *Cd209*[+] DC, *Cd209a* and *Flt3* enriched; *Xcr1* DC, *Xcr1* and *Itgae* enriched; and migratory DC, *Fscn1*, and *Cacnb3* enriched), monocytes (enriched genes: *Ly6c2*, *Chil3*, and *Ace*), neutrophils (enriched genes: *S100a8*, *S100a9*, and *Retnlg*), B cells (enriched genes: *Ms4a1*, *Cd79a*, and *Ly6d*), T cells (enriched genes: *Cd3e*, *Cd3d*, and *Lef1*), group 2 innate lymphoid cells (ILC2; enriched genes: *Rora*, *Cxcr6*, and *Gata3*), natural killer (NK) cells (enriched genes: *Nkg7*, *Klrb1c*, and *Gzma*), plasma cells (enriched genes: *Iglv1* and *Mzb1*), and mast cells (enriched genes: *Cma1* and *Kit*) (Fig. 1c and Supplementary Data 1).

In the steady-state, macrophages represented the largest cell population (58.7% of total CD45[+] cells) in the mouse hearts, followed by B cells (18.2%), NK cells (8.6%), T cells (4.3%), monocytes (3.7%), and neutrophils (3.1%) (Fig. 1d and Supplementary Table 1); the other cardiac leukocyte populations occupied less than 3% in the steady state. Interestingly, the proportion of macrophages drastically dropped at day 1 (24.9%) but then gradually recovered from day 3 (66.8%), reaching a peak on day 7 (84.0%) post-MI (Fig. 1d). On the contrary, the proportion of neutrophils increased steeply at day 1 (54.4%) and then decreased rapidly (14.8, 1.7, and 1.6% on days 3, 5, and 7, respectively), which was consistent with previous reports[12,15]. Monocytes were increased in the early phase of MI (days 1 and 3) and then decreased in the late phase of MI (days 5 and 7), while DCs were predominant at days 3 and 5. Lymphocytes (T cells, B cells, NK cells, and ILC2 cells), occupying 31.9% of total CD45[+] cells in the steady state, decreased at day 1 and did not reach

the steady-state values until day 7 post-MI. Collectively, scRNA-seq targeting cardiac leukocytes showed not only major cell populations but also minor populations, such as mast cells and DC subtypes; our data clearly show that the temporal dynamics after MI are different depending on the cell populations.

### Spatiotemporal profiles of immune cells after MI

To validate the above landscape of immune cells infiltrated into the infarcted area, we performed spatial transcriptome sequencing (ST-seq) using frozen samples from days 1, 3, 5, and 7 post-MI (Fig. 2a). Principal component analysis and unsupervised clustering identified four or five cell clusters per sample based on the differentially expressed genes (Fig. 2b). We then applied SPOTlight[16], a nonnegative matrix factorization (NMF)-based spatial deconvolution framework, to infer the cell-type composition of each spot (Fig. 2b). Notably, neutrophils highly infiltrated into the infarcted area on day 1 (41.2% on average) but their numbers were then decreased rapidly (18.7, 10.6, and 9.5% on days 3, 5, and 7, respectively), which was consistent with the scRNA-seq data (Figs. 2b and 2c). On the contrary, macrophages were dispersed across the whole heart rather than clustered in the infarcted area on day 1 (Fig. 2b). However, from day 3, macrophages infiltrated into the infarcted area, and their abundance peaked at late MI (days 5 and 7; 18.8% on average) (Fig. 2c). Monocytes and fibroblasts also infiltrated the infarcted area of the heart in a time-dependent manner (Supplementary Figs. 1–5). Other immune cell populations, such as B-cell and T-cell populations, were always dispersed across the entire mouse heart (not clustered in the infarcted area), with a very low abundance (Supplementary Figs. 2–5). Together with the scRNA-seq showing immune cell temporal dynamics after MI, ST-seq analysis with deconvolution algorithm further provided their spatial heterogeneity and spatiotemporal dynamics; our ST-seq data clearly showed that monocytes and neutrophils infiltrated into the infarcted area at early MI while macrophages and fibroblasts acted oppositely.

### The heterogeneity of monocytes and macrophages and their temporal dynamics after MI

scRNA-seq showed that macrophages represent the largest population of immune cells in the heart. For further insight regarding the dynamics of macrophage subtypes after MI, we performed a sub-clustering analysis in the context of scRNA-seq data, obtaining 16 sub-clusters (Fig. 3a). Detailed information of marker genes for each sub-cluster is provided in Supplementary Data 2. Three macrophage sub-clusters (clusters 1, 3, and 13) were predominant in the steady state (77.9% of the total monocyte/macrophages), thus we named them steady-state macrophage (SS-Mφ) clusters (Fig. 3b). In the SS-Mφ1 cluster (cluster 3), elevated expression of genes such as *Lyve1*, *F13a1*, *Cbr2*, *Cd163*, *Folr2*, and *Timd4* was observed (Supplementary Data 2 and Supplementary Figs. 6, 7); importantly, the increased expression of *Lyve1* and *F13a1* was previously reported in tissue-resident macrophages[17,18] and recent single-cell studies also suggested that the upregulation of *Folr2*, *Timd4*, and *Cbr2* is a marker of resident macrophages[9,19]. The SS-Mφ2 cluster (cluster 1) showed relatively higher expression of antigen-presentation-related genes, including *H2-Eb1*, *H2-Aa*, *H2-Ab1*, and *Cd74*, resembling the tissue-resident *Mhc-II* macrophage cluster suggested by Dick et al[9]. Additionally, the marker genes of both SS-Mφ1 and SS-Mφ2 were detected in the SS-Mφ3 cluster (cluster 13), indicating that these cells represent an intermediate state between those in the SS-Mφ1 and SS-Mφ2 clusters. Expectedly, *Ccr2*, a marker of monocyte-derived macrophages, was not expressed in the SS-Mφ1, whereas it was partially expressed in the SS-Mφ2 (Supplementary Fig. 7)[9]. Interestingly, the proportions of the SS-Mφ clusters dropped drastically after the induction of MI and were then gradually restored from day 3 (SS-Mφ2 and SS-Mφ3) or day 5 (SS-Mφ1) after MI (Fig. 3b). However, their proportion did not reach that in the steady-state until day 7 post-MI.

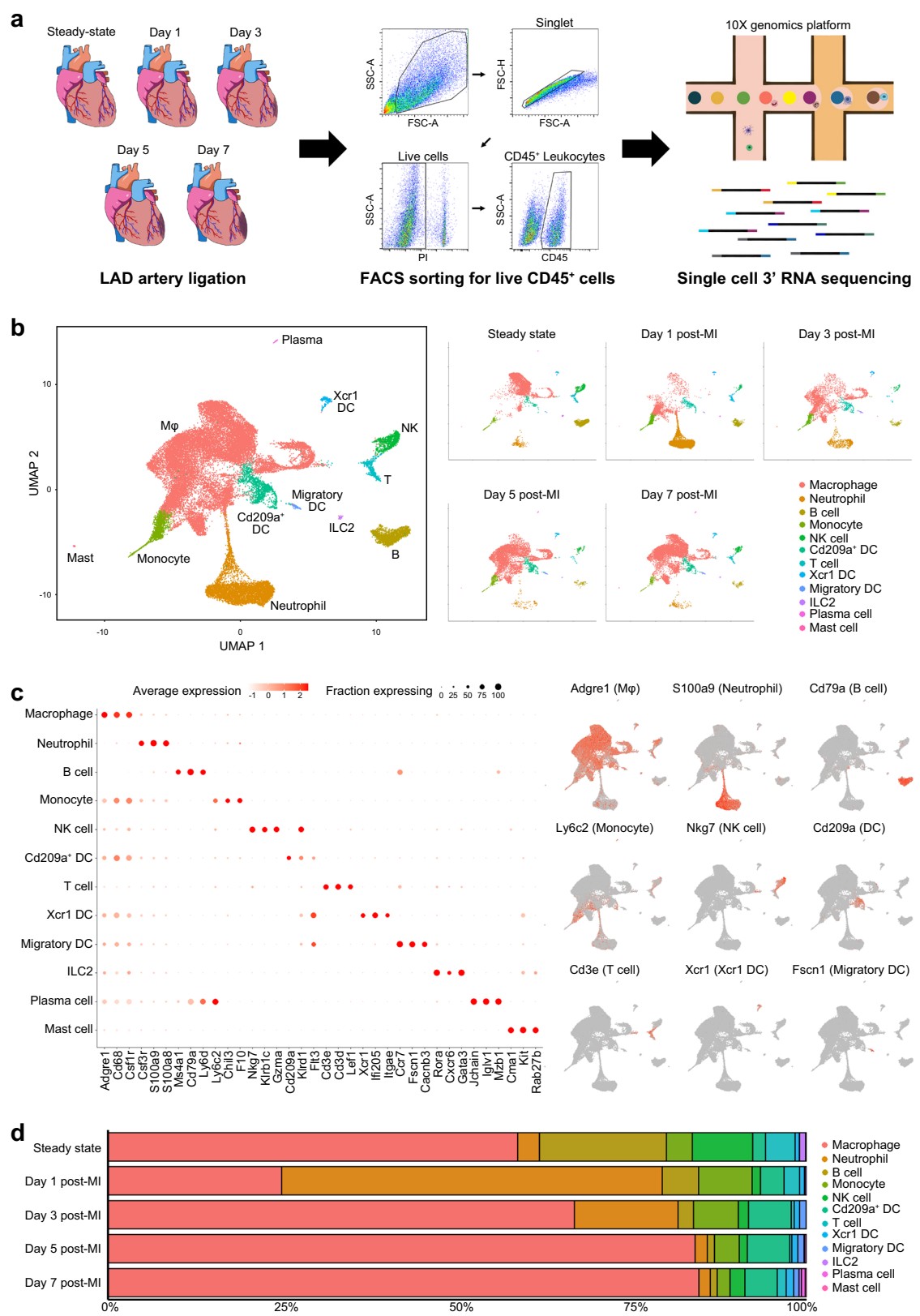

Interestingly, in the early phase of MI (days 1 and 3), monocytes and two macrophage clusters were predominant (clusters 4, 10, and 14), occupying 59.7% of the total monocyte/macrophage cells at day 1 (termed Early-Mφ clusters). Since cluster 14 showed the upregulation of *Chil3*, *Plac8*, *Ccr2*, and *Ly6c2*, it was defined as classical Ly6c2$^{hi}$ monocytes (Supplementary Data 2 and Supplementary Fig. 6); Ly6c2$^{hi}$

monocytes peaked one day post-MI and then quickly decreased (Fig. 3b). Additionally, Early-Mφ1 (cluster 10) and Early-Mφ2 (cluster 4) expressed not only *Cd68*, *Fcgr1*, and *Itgam* (canonical macrophage markers) but also *Ccr2*, suggesting that these populations were composed of monocyte-derived macrophages (Supplementary Fig. 6). Like monocytes, these two Early-Mφ populations peaked at day 1 and then

**Fig. 1 | Time-dependent immune cell dynamics after myocardial infarction.**
**a** Graphical representation of the experimental setup. Experimental MI was induced in mice by the permanent ligation of the proximal portion of the left anterior descending artery (LAD). Samples were isolated from mouse hearts at days 0, 1, 3, 5, and 7 post-MI. Tissues were enzymatically digested and live CD45+ cells were FACS-sorted and loaded for scRNA-seq. **b** Two-dimensional uniform manifold approximation and projection (UMAP) visualization of the 33,977 cardiac CD45+ cells identified 12 broad cell types after unsupervised clustering. Each point represents a single cell; cell types are color-coded. **c** Left: dot plot of well-known cell-type-specific marker genes per cell type. The dot intensity (from white to red) represents the average expression value of all cells per cell type and the dot size represents the proportion of cells expressing the genes. Right: feature plot representing the expression levels of the selected cell-type-specific marker genes. **d** Bar plot representing the proportions of cells in each of the 12 broad cell types according to the time-point after MI. Source data are provided in the Source Data file.

decreased until day 7 post-MI (Fig. 3b). Interestingly, the expression of *Ccr2* was inversely correlated between Ly6c2hi monocytes, and Early-Mφ1 and Early-Mφ2 cells ($R^2 = 0.103$, $P = 4.2 \times 10^{-84}$; Supplementary Fig. 8), suggesting that circulating monocytes are recruited into the infarcted areas, where they differentiate into macrophages. In addition, a Ly6clow population as non-classical monocytes (cluster 16, enriched genes: *Ace*, *Ear2*, and *Itgal*) was identified at a lower level and its proportion was not markedly changed over time after MI (Fig. 3b and Supplementary Fig. 6).

Additionally, in the late phase of MI (days 5 and 7), four macrophage clusters were predominant (59% of the total monocyte/macrophage cells), particularly at day 7; two of which were steady-state macrophages (SS-Mφ1 and SS-Mφ2) and the other two populations (cluster 2 and 5) were termed Late-Mφ clusters. Late-Mφ1 (cluster 2) cells showed a relatively higher expression of *Apoe*, *Fcrls*, *Rgs10*, and *Adgre1*, while Late-Mφ2 (cluster 5) cells showed a higher expression of *Trem2*, *Gpnmb*, *Fabp5*, and *Spp1* as well as the cardiac repair gene *Timp2* (Supplementary Figs. 6, 7)[20]. These two Late-Mφ populations gradually increased from day 3, peaking 7 days post-MI (Fig. 3b).

We also identified three transient macrophage populations (peaking 3 or 5 days after MI), two macrophage populations enriched in interferon-stimulated genes, and two proliferating populations, albeit their proportions were low (Fig. 3b). *Saa3*, *Fn1*, and *Ltc4s* genes were enriched in transient-Mφ1 (cluster 8) cells, *Fabp5*, *Spp1*, and *Gpnmb* genes were enriched in Transient-Mφ2 (cluster 7) cells, and *Hmox1*, *Prdx1*, and *Gclm* genes were enriched in Transient-Mφ3 (cluster 12) cells. The significantly enriched genes in Transient-Mφ populations were similar to those in Late-Mφ2 cells; however, the expression levels of *Trem2* were relatively lower in Transient-Mφ cells (Supplementary Figs. 6, 7). Additionally, many genes associated with interferon signatures such as *Irf7*, *Isg15*, and *Ifit2* were enriched in clusters 6 and 15, which were therefore defined as IFN-Mφ (Fig. 3a, b); the proportions of the IFN-Mφ clusters did not significantly change over time after MI (Fig. 3b). Cluster 15 showed a higher level of *Ccr2* expression than cluster 6, indicating that this cell population may be derived from monocytes (Supplementary Fig. 6). The two clusters corresponding to proliferating macrophages were defined as the S phase (cluster 11) and G2/M phase (cluster 9) populations (Fig. 3a, b and Supplementary Fig. 9). Overall, the sub-clustering analysis clarified macrophage heterogeneity and allowed disclosure of the temporal dynamics of macrophages/monocytes in the heart. Importantly, these data indicate that Late-Mφ populations, as well as SS-Mφ populations, may have central functions in attenuation of the inflammatory response post-MI and the promotion of tissue repair.

## Single-cell trajectories in the infarcted heart

To better understand the process of monocyte/macrophage infiltration into the infarcted area, we performed a trajectory analysis using the Monocle R package[21]. Ly6c2hi monocytes share several properties with human CD16−CD14+ monocytes, and it is well known that they enter the circulation, contribute to excessive monocytosis, preferentially accumulate in lesions, and differentiate into macrophages after MI[6]. When Ly6c2hi monocytes were set as the root of the trajectory, the pseudo-time of cells increased in the order of Early-Mφ, Transient-Mφ, and Late-Mφ, which were ranked in a similar temporal manner (Fig. 3c). However, Transient-Mφ1 and IFN-Mφ populations

diverged at a relatively early pseudo-time, and Transient-Mφ1 was not linked to Late-Mφ. The selected genes that are specifically expressed in each cell population were then plotted to track changes across pseudo-times (Fig. 3d). The expression levels of Early-Mφ specific genes, such as *Ccr2*, *Chil3*, and *Clec4e*, were upregulated in Ly6c2hi monocytes/Early-Mφ and then gradually decreased in Transient-Mφ, reaching a low level in Late-Mφ cells. On the contrary, Late-Mφ-specific genes, such as *Trem2*, *Rgs10*, and *Fcrls*, were downregulated in Ly6c2hi monocytes/Early-Mφ and then gradually upregulated in Transient-Mφ and Late-Mφ cells (Fig. 3d). *Trem2* was specifically upregulated in Late-Mφ cells, while *Folr2*, *Lyve1*, and *Mgl2* expressions were very low. Importantly, when we checked the expression of candidate genes and the proportion of macrophage subsets in injured heart tissues using ST-seq, marker genes and proportions for each subset were enriched in the infarcted area of the heart in a time-dependent manner, while SS-Mφ1 cells were dispersed in the heart at all time points after MI (Fig. 3e and Supplementary Figs. 10, 11).

## Neutrophil and DC subpopulations

Neutrophils, previously reported as the first immune cells recruited into the infarcted area[15], were also the largest cell population one day post-MI (Fig. 1d). Using the 5,135 neutrophils (steady-state: 204 cells; day 1: 4009 cells; day 3: 660 cells; day 5: 126 cells; day 7: 136 cells), we further identified five distinct neutrophil sub-clusters (Fig. 4a). When we calculated the proportion of each cluster in relation to the total neutrophil counts, cluster 1 (enriched genes: *Fpr1*, *S100a9*, and *Lcn2*) and cluster 3 (enriched genes: *Ifitm2*, *Pglyrp1*, and *Slpi*) were predominant in the steady state (31.9 and 63.7%, respectively; Fig. 4b). However, the proportion of cluster 3 notably decreased early after MI (day 1; 11.9%) and then gradually increased to 36.0% at day 7, while cluster 1 did not show any evident trend. Conversely, the proportion of cluster 2 (enriched genes: *Icam1*, *Tnf*, and *Siglecf*) was rarely presented in the steady state (0.5%) but notably increased at day 1 (21.9%) and peaked at day 3 (46.8%) post-MI. This is consistent with a previous report suggesting that neutrophils in the infarcted heart are characterized by the acquisition of a SiglecFhi signature[12]. Cluster 4 was enriched in genes associated with interferon signatures such as *Ifit1*, *Rsad2*, and *Isg15*[12], and cluster 5 was characterized by the upregulation of endoplasmic reticulum heat-shock proteins such as *Hspa5*, *Dnajb9*, *Manf*, and *Hyou1* (Fig. 4a)[22]. The gene expression levels of typical cluster-specific genes are provided in Fig. 4c and Supplementary Data 3. Notably, proportions of both clusters 4 and 5 peaked one day after MI.

Regarding DCs, six distinct sub-clusters were identified as per the sub-clustering analysis (steady-state: 160 cells; day 1: 309 cells; day 3: 349 cells; day 5: 568 cells; day 7: 535 cells; Fig. 4d). Three of them were predominant before MI (clusters 2, 3, and 4; 93.8% of the total DCs) but their proportions gradually decreased to 44.5% 5 days post-MI (Fig. 4e). In cluster 2, genes such as *Tgfbi* and *Ccr2* were upregulated, while cluster 3 showed a relatively higher expression of DC-SIGN (*Cd209a*) and MHC II molecules (*H2-DMa* and *H2-Aa*) (Fig. 4d). Additionally, cluster 4 showed a distinct transcriptional state with a high expression of *Xcr1* and *Clec9a*, suggesting that this population is composed of cDC1[23]. On the other hand, macrophage (*Ms4a7* and *Fcrls*) and migratory DC (*Fscn1* and *Ccr7*) signatures were detected in clusters 1 and 5,

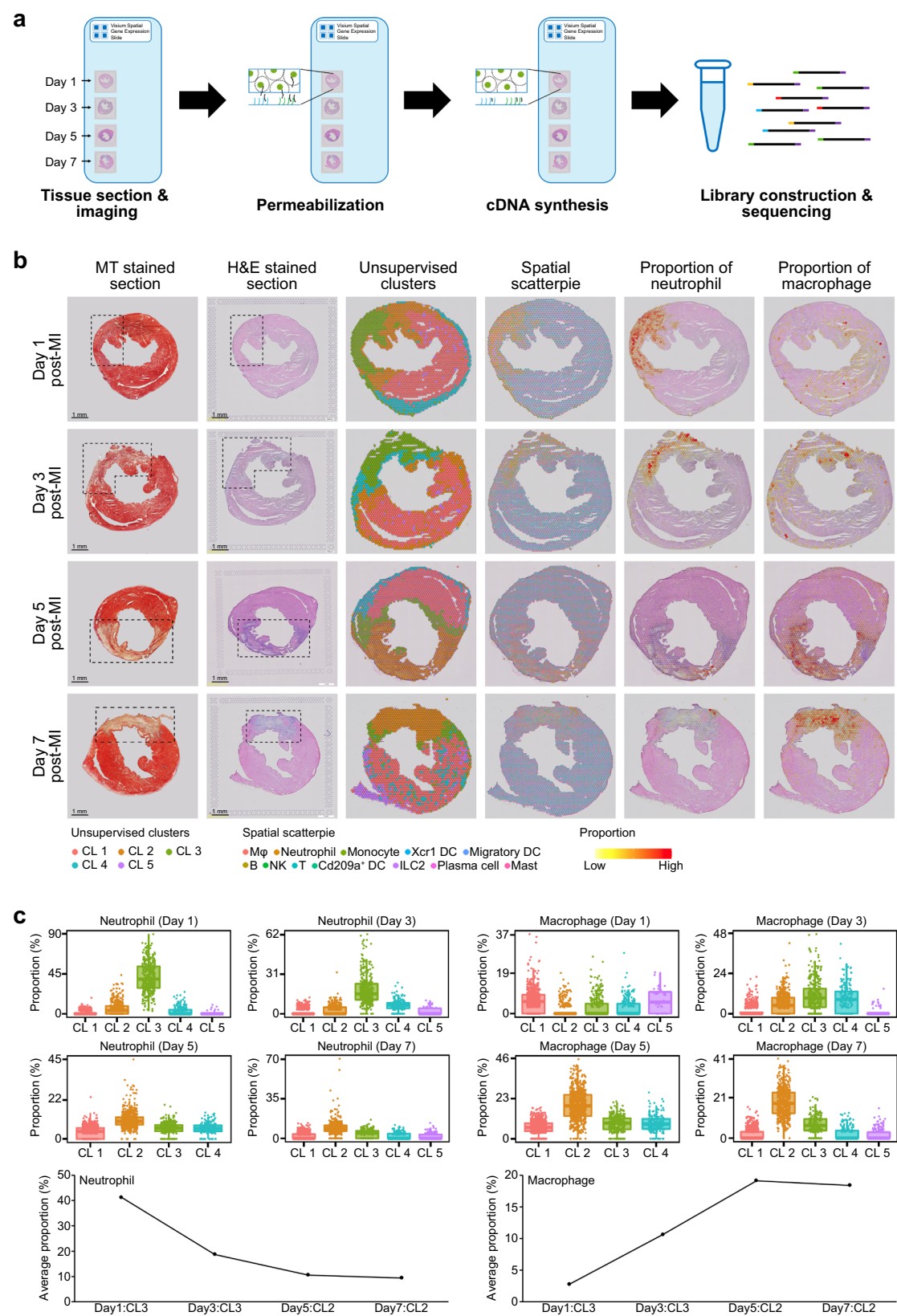

respectively (Fig. 4d); their abundance was relatively higher in the late phase than that in the early phase of MI (Fig. 4e). Finally, cluster 6 was characterized by the upregulation of plasmacytoid DC marker genes such as *Siglech* and *Bst2* (Fig. 4d). The gene expression levels of typical cluster-specific genes are provided in Fig. 4f and Supplementary Data 4.

**The expression of Trem2 after MI**

Previous studies showed the increased expression of *Trem2* in mouse hearts during atherosclerosis progression and regression[19,24]. In this study, immunohistochemistry also showed a gradual increase in the expression of Trem2 over time in the infarcted area after the induction of MI, which was almost absent in the steady state (Fig. 5a). Consistent

**Fig. 2 | Spatial transcriptome sequencing (ST-seq) of mouse hearts after MI.**
**a** Schematic workflow of ST-seq. Frozen heart sections were placed onto Visium Spatial Gene Expression slides and then permeabilized; cDNAs were synthesized on the slides. **b** The first and second column represents Masson's trichrome- and H&E-stained images showing the infarcted area (dotted box). In the third column, each spot contains 1–10 cells on average, colored according to the defined clusters using Seurat. The fourth column represents the spatial scatter pie plot which shows the proportions of the immune cells. The proportions were deconvoluted from the scRNA-seq data using the SPOTlight algorithm. The fifth and sixth columns represent the neutrophil and macrophage proportion, respectively. Yellow and red indicate lower and higher proportions, respectively. Results are representative of four different samples. **c** The proportion of neutrophils and macrophages among total immune cells infiltrated into heart tissue according to the time-point after MI. Each cluster from the time-point after MI has the same number of spots (day 1 post-MI: 1803 spots; day 3 post-MI: 1916 spots; day 5 post-MI: 1870 spots; day 7 post-MI: 1973 spots). The lower whisker, lower hinge, box center, upper hinge, and upper whisker represent the minimum, lower quartile, median, upper quartile, and maximum calculated without outlier values which are more than 1.5× interquartile range of the lower and upper quartiles. The proportion of all immune cell types are illustrated in Supplementary Figs. 2–5. Source data are provided in the Source Data file.

with scRNA-seq, ST-seq, and immunohistochemistry results, western blotting showed that the expression of full-length Trem2 (~32 kDa) steadily increased, peaking 5 days post-MI and decreasing thereafter (Fig. 5b). Interestingly, the expression of the soluble form of Trem2 (sTrem2, ~18 kDa) increased after day 3 and peaked on day 7. The expression of total Trem2 (full-length Trem2 and sTrem2) peaked on day 5 (Fig. 5b). Therefore, we decided to investigate the main source of Trem2 in the heart; to investigate whether Trem2hi macrophages are the major cell population responsible for the Trem2 expression pattern after MI, we performed a co-localization assay. Importantly, CD68+ macrophages increased markedly in the infarcted area and the Trem2 signals overlapped with those of CD68+, suggesting that Trem2 overexpression occurs mostly in macrophages (Fig. 5c). We also evaluated the secretion levels of sTrem2 in Trem2hi and Trem2low macrophages from days 3 and 5 post-MI and identified that sTrem2 was secreted only in Trem2hi macrophages (Supplementary Fig. 12). Of note, the secretion level of sTrem2 on day 5 was significantly higher than that on day 3. Collectively, these results indicate that Trem2 is highly expressed by macrophages in the infarcted heart and sTrem2 is exclusively secreted from Trem2hi macrophages.

### Trem2hi macrophages accumulating in injured hearts late after MI are anti-inflammatory

We further characterized Trem2hi macrophages in our MI mouse model. Single-cell suspensions obtained from the heart tissues were analyzed by flow cytometry. To specifically collect Trem2hi macrophage subsets, we sorted the immune cells based on CD45 expression, and then the macrophage populations were selected using F4/80 antibody. Ccr2hi macrophages and Trem2hi macrophages, as major subsets in the early and late phases of MI, were then collected (Supplementary Fig. 13). The proportion of Ccr2hi macrophages rapidly increased at day 1 post-MI and then declined over time (Fig. 5d), suggesting that monocyte-derived macrophages were recruited to the infarcted areas at early MI, which was consistent with a sub-clustering analysis of scRNA-seq data. In contrast, the proportion of Trem2hi macrophages began to increase at day 3 and peaked 5 days after MI, indicating a specific dominance in late MI in the infarcted heart (Fig. 5e, f). Additionally, the mean fluorescence intensity of Trem2 in F4/80+ macrophages gradually increased after MI and peaked at day 5 (Fig. 5g).

To investigate the polarizing status of the Ccr2hi and Trem2hi macrophage subsets, we analyzed the expression levels of Arginase 1 (Arg1). Significantly higher levels of Arg1 were observed in Trem2hi macrophages than in Ccr2hi macrophages (Fig. 5h). In the quantitative reverse transcription-PCR analysis, Ccr2hi macrophages exhibited higher expression levels of pro-inflammatory genes (cytokines: *Il1b* and *Il6*, chemokine: *Ccl2*, and chemokine receptors: *Cxcr3* and *Cxcr7*), whereas Trem2hi macrophages expressed higher levels of anti-inflammatory genes (cytokines: *Il10* and *Tgfb1*, chemokine receptor: *Cx3cr1*, and enzyme: *Alox15*) (Fig. 5i, j). Of note, Trem2hi macrophages also exhibited higher expression level of osteopontin (*Spp1*), which is related to pro-fibrotic potential in regulating post-MI LV remodeling[25,26]. These results indicate that Trem2hi macrophages may

have characteristics of anti-inflammatory macrophages. We further investigated the sTrem2 effects on macrophage polarization using thioglycolate-elicited peritoneal macrophages. sTrem2 treated pro-inflammatory macrophages showed decreased *Il12b* and *Nos2* expression (pro-inflammatory markers) and increased *Arg1* expression, suggesting that sTrem2 affects macrophage polarization (Supplementary Fig. 14). Collectively, these data demonstrate that Trem2hi macrophages are dominant in the late phase of MI and possess anti-inflammatory characteristics.

### Improvement of left ventricular remodeling after in vivo injection of soluble Trem2

To verify the in vivo effects of sTrem2 on the remodeling after MI, we generated an injectable gelatin hydrogel (GH) mixed with sTrem2 as described previously[27] and compared the changes in infarct size and echocardiographic LV systolic function among three groups of mice with MI: Those treated with PBS, those treated with GH alone, and those treated with GH mixed with sTrem2 (sTrem2-GH, 12 μg) (Fig. 6a). Importantly, based on the mechanical strength of GH, the crosslinking densities of GH alone and sTrem2-GH were fixed at 1.5 kPa. In vitro, the sTrem2 release profile test from sTrem2-GH showed that 70% of sTrem2 was released out of sTrem2-GH at day 1 and approximately 90% was cumulatively released over 7 days at a slow pace (Supplementary Fig. 15). Importantly, 28 days post-MI, echocardiography showed that the mice treated with sTrem2-GH had significantly higher left ventricular ejection fraction (LVEF), fractional shortening (FS), and lower end-systolic volume (ESV) than the mice treated with PBS or GH alone (Fig. 6c, d, Supplementary Table 2 and Movies 1–3). Histologically, the mice treated with sTrem2-GH showed less dilated and well-remodeled LV with thicker infarcted walls and lower fibrosis (Fig. 6b and Supplementary Fig. 16). The infarct size was also significantly smaller in sTrem2-GH-treated MI mice (26.0%) than that in PBS-(48.4%), and GH-treated (38.0%) mice (Fig. 6e) and an increased survival rate was observed in the mice treated with sTrem2-GH compared with other groups (Fig. 6f). Altogether, these results indicate that sTrem2 promotes the functional and structural improvements of infarcted hearts in vivo.

### Discussion

In this study, we attempted to identify the heterogeneity and spatio-temporal dynamics of leukocytes at a single-cell level in the mouse heart after the induction of MI using longitudinal scRNA-seq and ST-seq. Our data support four major conclusions: First, leukocytes are heterogeneous in the heart tissues with respect to not only inter-cell (12 broad cell populations) but also intra-cell (16 monocytes/macrophage subsets, 5 neutrophil subsets, and 6 DC subsets) populations; second, leukocyte populations after the initiation of MI are remarkably dynamic, changing depending on the cell type or cell subpopulations; third, the single-cell trajectory analysis supports a sequential differentiation from Ly6c2hi monocytes to Late-Mφ rather than the obsolete dichotomous M1-M2 paradigm; fourth, in the late phase of MI, the number of Trem2hi macrophages abundantly expressing anti-inflammatory signature genes is significantly increased in the

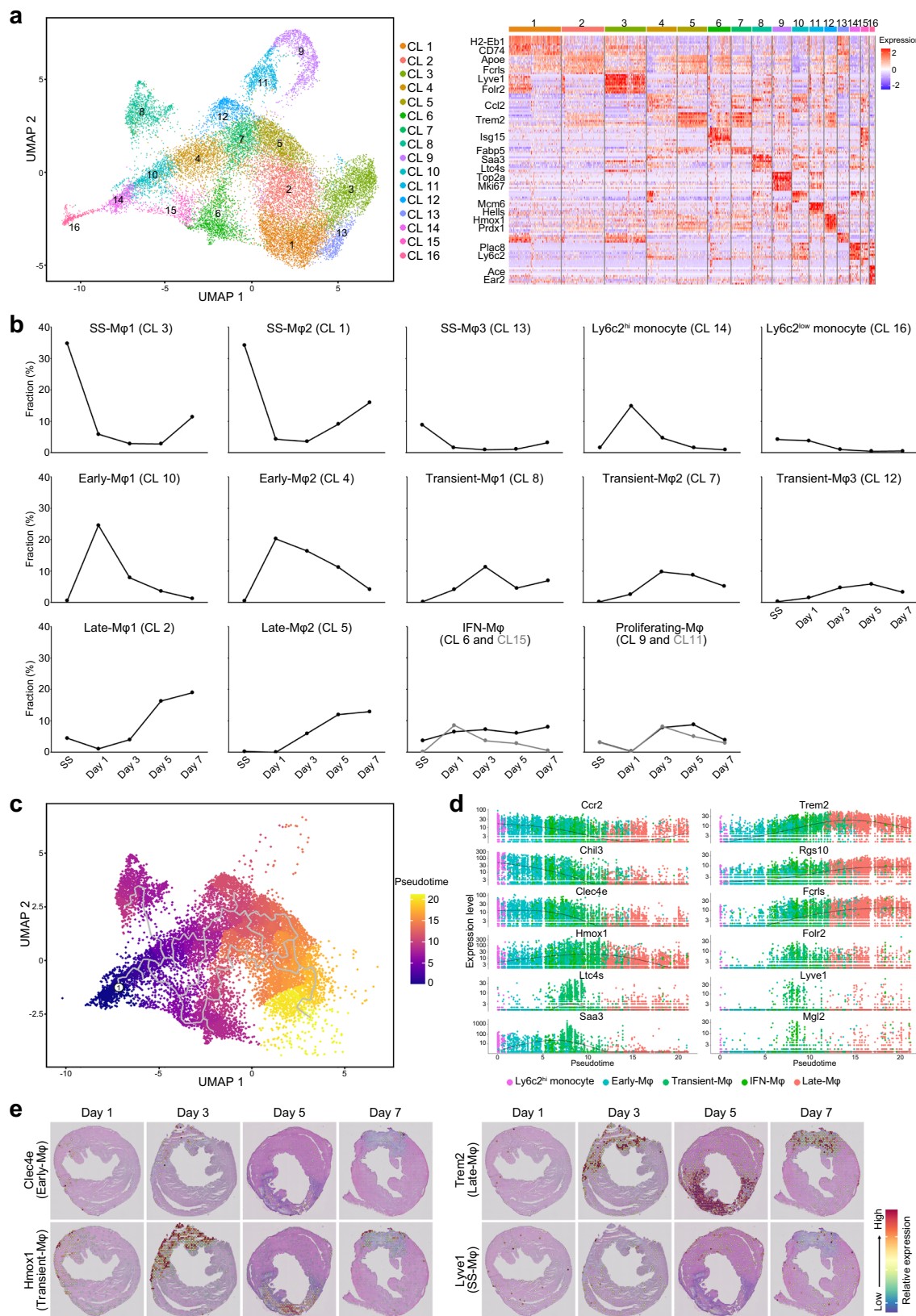

infarcted heart tissues and the soluble form of Trem2, the expression of which increased after the peak of full-length Trem2 expression, significantly improves remodeling and cardiac function in the infarcted heart. Altogether, our results demonstrate that the cellular components of the immune system in the mouse heart are heterogeneous and have plasticity in response to ischemic injury. These findings can provide a rich resource to explore the molecular targets and underlying mechanisms of tissue repair after MI and thus may guide the development of novel therapeutic strategies aimed at enhancing myocardial repair and regeneration.

Macrophages are composed of several heterogeneous subpopulations and recent investigations adopting single-cell

**Fig. 3 | Monocyte/macrophage cell subsets. a** The UMAP visualization of the 21,533 cardiac monocytes/macrophages identified 16 subsets (left panel). Each point represents a single cell, colored according to the sub-cluster assigned. In the right panel, the heatmap shows the top 10 most differentially expressed genes in each sub-cluster. Blue and red indicate lower and higher expression, respectively. The typical markers strongly and specifically associated with each sub-cluster are shown on the left. **b** The proportion of each sub-cluster among total monocytes/macrophages according to the time-point after MI. **c** Pseudo-time trajectory as per the pseudo-time algorithm. Pseudo-time analysis is an approach used to investigate the path and progress of individual cells undergoing differentiation. Ly6c2[hi]

monocytes were set as the root of the trajectory. The scale indicates the temporal status, from dark blue (recruited Ly6c2[hi] monocytes) to yellow (Late macrophages). **d** Spline plots showing the expression of typical markers associated with each cluster across the pseudo-time (Ly6c2[hi] monocytes: 632 cells; Early-Mφ: 2838 cells; Transient-Mφ: 3146 cells; IFN-Mφ: 1856 cells; Late-Mφ: 4380 cells). **e** Gene expression levels of *Clec4e*, *Hmox1*, *Trem2*, and *Lyve1* as per ST-seq. Blue and red indicate lower and higher expression, respectively. The scale bar is marked on the H&E stained section in Fig. 2b. Results are representative of four different samples. Day 1, day 1 post-MI; Day 3, day 3 post-MI; Day 5, day 5 post-MI; Day 7, day 7 post-MI. Source data for Figs. 3b and 3d are provided in the Source Data file.

technologies have already implicated some macrophage subsets in injured hearts[9–11,24,28]. However, the whole picture concerning macrophage heterogeneity as well as the functions of each subset in the repair of infarcted tissues following MI are still unknown. Therefore, we explored the subpopulations of the monocytes/macrophages defined by scRNA-seq and identified 16 subsets. These subsets included Lyve1[hi] and Mhc-II[hi] steady-state macrophages, pro-inflammatory monocytes and macrophages identified 1 and 3 days post-MI (Early-Mφs), macrophages identified 5 and 7 days post-MI (Late-Mφs) as well as several minor populations (non-classical monocytes, IFN-Mφs, and proliferating-Mφs); they were enriched in the infarcted area at different time-points after MI. Notably, SS-Mφ populations decreased after MI and were restored on days 5–7. Considering the previous study, which reported that resident cardiac CCR2⁻TIMD4⁺ Mφ (SS-Mφ1 in this study) were maintained independently of monocytes, while CCR2⁺/⁻TIMD4⁻ MHCII[hi] Mφ (SS-Mφ2 in this study) were fully or partially replaced by monocyte over time[9], SS-Mφ1 might be restored due to proliferation of the remaining resident cells, while SS-Mφ2 might be replaced. Further in-depth analysis using fate-mapping will provide their conclusive origin. We also identified a macrophage subset, Trem2[hi] macrophages with anti-inflammatory characteristics, specifically dominant in the late phase of MI in the infarcted heart. Pseudo-time analysis is an approach used to investigate the path and progress of individual cells undergoing differentiation[21]. Although our inferred trajectories harbored a limitation that Ly6c2[hi] monocytes were artificially designated into root state with simple tree-structure models[29], the pseudo-time analysis of monocytes/macrophages matched the order of sampling time-points after MI, supporting a sequential differentiation from Ly6c2[hi] monocytes to Late-Mφ rather than the conventional dichotomous M1–M2 paradigm.

Trem2 is a member of the immunoglobulin superfamily that propagates activation signaling via the adaptor proteins DAP10 and DAP12[30]. Previous studies have reported that Trem2 signaling induces significant changes in cellular phenotypes and functions, such as the induction of phagocytosis, restriction of inflammation, and promotion of macrophage survival under tissue damage conditions[30–34]. Moreover, previous scRNA-seq studies showed the Trem2[hi] macrophage population in the mouse heart during atherosclerosis progression and regression[19,24]. In addition, recent studies suggested that Trem2 is a potential therapeutic target to modulate immunosuppressive tumor-associated macrophages[35,36]. However, the infiltration behavior and impact of Trem2[hi] macrophages on MI healing have never been investigated. Herein, we showed the increased infiltration of Trem2[hi] macrophages specifically in the late phase of MI, not only using scRNA-seq, but also via orthogonal analyses such as immunohistochemistry, immunocytochemistry, western blotting, flow cytometry, and qRT-PCR. Of note, a higher expression of anti-inflammatory signature genes was observed in Trem2[hi] macrophages, suggesting anti-inflammatory macrophage characteristics. We used an antibody against the C-terminal Trem2 for western blotting analysis, which could detect both full-length and sTrem2. sTrem2 was reported to be detected in human cerebrospinal fluid[37], and the estimated molecular weight of

sTrem2 is ~17 kDa, which matches the size of the lower band (18 kDa) observed in this study. Curiously, while the expression of full-length Trem2 peaked at day 5 post-MI, that of sTrem2 was elevated afterward, suggesting that sTrem2 in the injured heart was generated as a proteolytic by-product of Trem2 rather than via alternative splicing[38,39]. Importantly, when we used an injectable gelatin-based hydrogel, which provided physical support for the infarcted myocardium and an appropriate microenvironment for simultaneously delivering biomolecules[27], the injection of sTrem2 with GH into the peri-infarct area of MI in mice resulted in higher LVEF/FS, lower ESV, and morphologically less dilated and well-remodeled LV than those in control animals, supporting the inflammation-modulating function and anti-remodeling effect of Trem2 in vivo. Further studies are, however, needed to systematically evaluate the Trem2[hi] macrophage origin, the cells affected by sTrem2, and the related intracellular signaling pathways, to enable a better understanding of its regenerative effect in infarcted hearts, before conducting human clinical trials.

In addition to macrophages, other leukocyte subpopulations are also worth noting in the context of post-MI immune responses. The neutrophil sub-clusters 2 and 3 were relatively abundant at days 3 and 5 post-MI, and their transcriptional states were coherent with those of Siglecf[hi] and Siglecf[low] neutrophils reported by Vafadarnejad et al[12]. Importantly, both Siglecf[hi] and Siglecf[low] neutrophils have been reported to be major subsets at days 3 and 5 post-MI[12]. Sub-cluster 2 (enriched genes: *Icam1* and *Siglecf*) was rarely present in the steady-state (1 of 204 neutrophils, 0.5%) but abundant after the induction of MI. *Icam1* can enhance effector functions such as ROS production and phagocytosis;[40] importantly, such functions were observed in SiglecF[hi]ICAM[hi] compared with SiglecF[low]ICAM[low] neutrophils[12]. Therefore, the transcriptional signature of the neutrophil sub-cluster 2 might be acquired after MI and have an important function in the modulation of immune responses. Moreover, among the DC subsets, the abundance of sub-clusters 1 and 5 (migratory signature) was relatively higher in the late phase after MI. Although we were unable to discover the biological functions in detail herein, previous studies have reported that migratory DCs characterized by *Fscn1* expression are associated with the dampening of immune responses[41,42]. Nevertheless, further in-depth exploration of a sufficient number of neutrophils and DCs during MI progression would be helpful to discover novel therapeutic targets to develop innovative treatments for MI.

We further explored the longitudinal spatial transcriptome profiles of the ischemic injured mouse heart, from day 1 to day 7 post-MI, using ST-seq. ST-seq incorporates unbiased gene expression profiles for tissue sections with morphological context, offering a much deeper understanding of immune cell infiltration[43]. Although ST-seq used in this study does not provide spatial expression profiles at true single-cell resolution (1–10 cell resolution on average per spot), ST-seq analysis with deconvolution algorithm indicated that the proportions of monocytes, neutrophils, and macrophages increased or decreased in a time-dependent manner, which was consistent with previous reports[15]. Moreover, they were concentrated in the infarcted area, indicating that these cell populations were recruited to the injured heart tissues after MI. Interestingly, among the macrophage subsets defined by scRNA-Seq, Early-Mφ, Transient-Mφ, and Late-Mφ populations were enriched

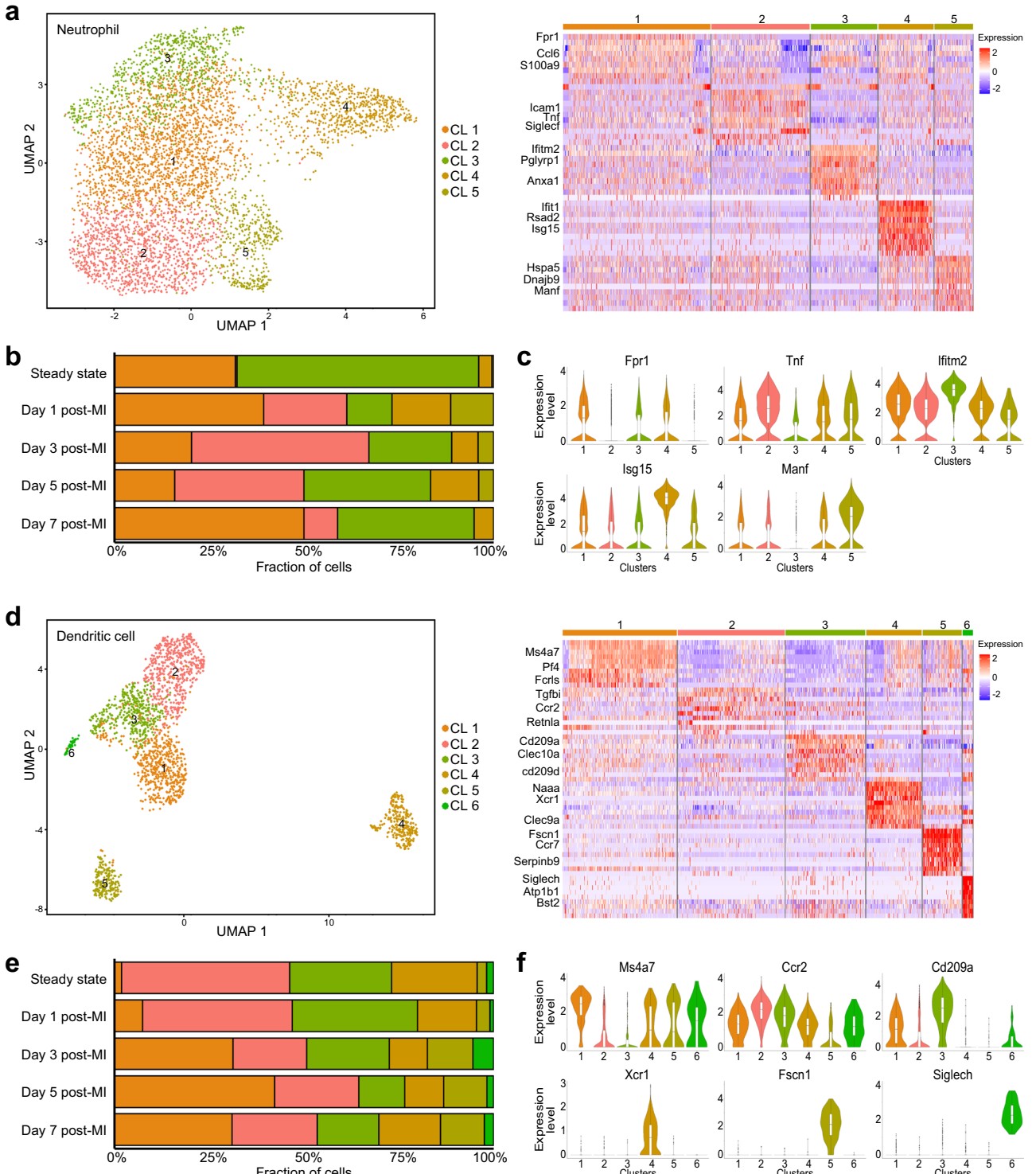

**Fig. 4 | Neutrophil and dendritic cell subsets. a** The UMAP visualization of 5135 neutrophils identified five distinct subsets (left panel). Each point represents a single cell, colored according to the sub-cluster assigned. In the right panel, the heatmap shows the top 10 most differentially expressed genes in each sub-cluster. Blue and red indicate lower and higher expression, respectively. Typical markers strongly and specifically associated with each sub-cluster are shown on the left. **b** Bar plot showing the proportions of cells in each of the 5 sub-clusters according to the time-point after MI. **c** Violin plots of the expression of *Fpr1, Tnf1, Ifitm2, Isg15*, and *Manf* in neutrophil sub-clusters (Cluster 1: 1864 cells; Cluster 2: 1243 cells; Cluster 3: 844 cells; Cluster 4: 698 cells; Cluster 5: 486 cells). **d** The UMAP visualization of 1921 dendritic cells identified 6 distinct subsets (left panel). Each point represents a single cell, colored according to the sub-cluster assigned. In the right

panel, the heatmap shows the top 10 most differentially expressed genes in each sub-cluster. Blue and red indicate lower and higher expression, respectively. Typical markers strongly and specifically associated with each sub-cluster are shown on the left. **e** Bar plot showing the proportions of cells in each of the 6 sub-clusters according to the time-point after MI. **f** Violin plots of the expression of *Ms4a7, Ccr2, Cd209a, Xcr1, Fscn1,* and *Siglech* in DC sub-clusters. (Cluster 1: 541 cells; Cluster 2: 507 cells; Cluster 3: 377 cells; Cluster 4: 264 cells; Cluster 5: 184 cells; Cluster 6: 48 cells). Regarding the box polts of **c** and **f**, the lower whisker, lower hinge, box center, upper hinge, and upper whisker represent the minimum, lower quartile, median, upper quartile, and maximum calculated without outlier values which are more than 1.5× interquartile range of the lower and upper quartiles. Source data for **b, c** and **e, f** are provided in the Source Data file.

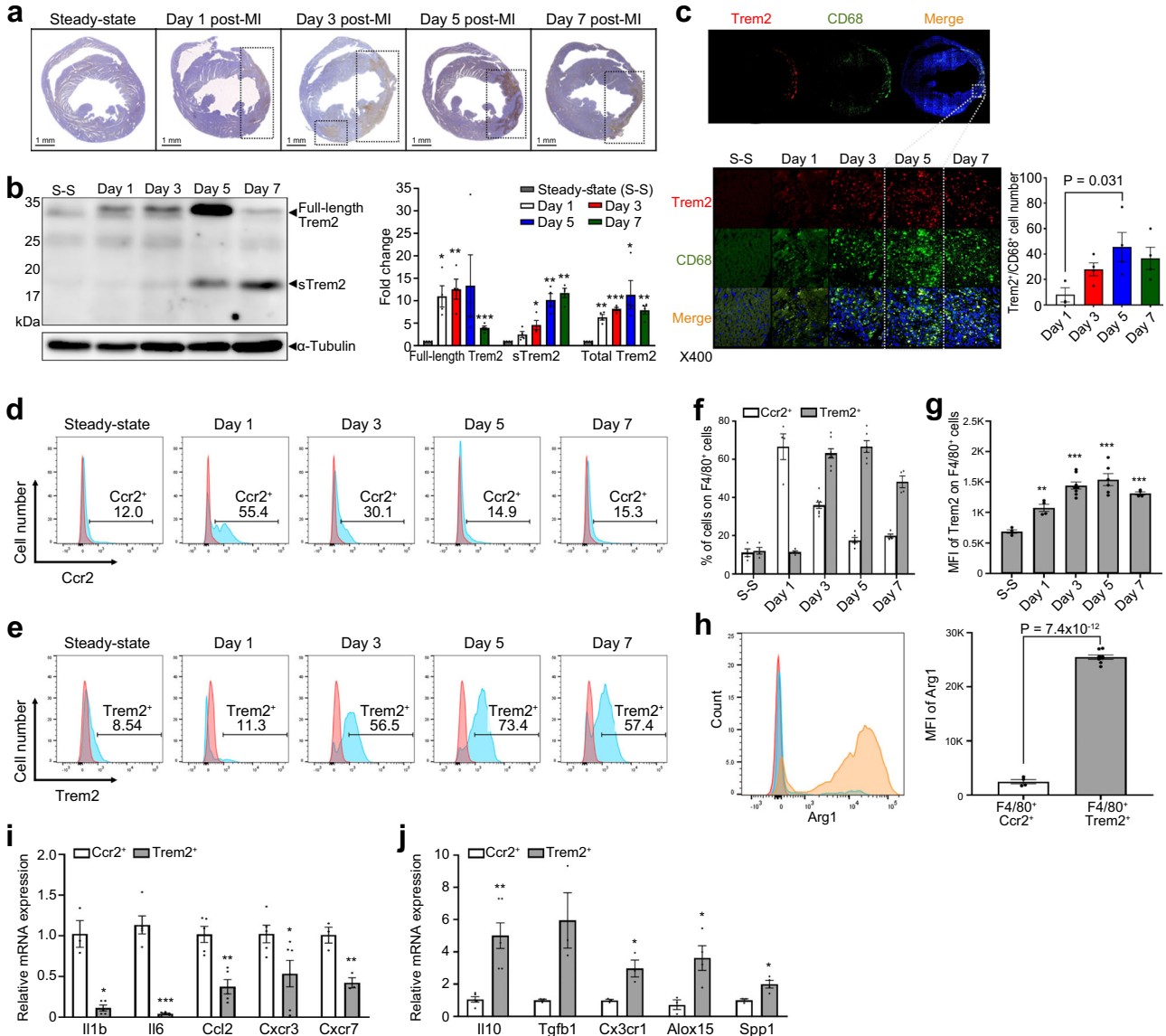

**Fig. 5 | Orthogonal validation of the expression of Trem2 in the heart after MI.**
**a** Immunohistochemistry showed a gradual increase in the expression of Trem2 over time in the infarcted area after the induction of MI, almost absent in the steady-state. Results are representative of five different samples. **b** Western blot targeting Trem2 showed two bands: a 32 kDa sized one corresponding to the full-length Trem2 protein and an 18 kDa sized one corresponding to soluble Trem2 (sTrem2); 50 µg of infarcted heart tissue proteins were loaded onto each lane. Quantitative comparison of the full-length, soluble form, and total Trem2 expression, respectively ($n = 4$ for each group). Unpaired two-tailed t-test was used to determine the statistical significance. *$P < 0.05$, **$P < 0.01$, ***$P < 0.001$. **c** Co-localization assay for Trem2 and the macrophage marker CD68 in the infarcted area according to the time-point after MI. Cells were stained with the specific antibodies anti-Trem2 (red) and anti-CD68 (green). Nuclei were counterstained with DAPI (blue). Quantification of Trem2+/CD68+ cells in the infarcted area ($n = 3$ for day 1 post-MI; $n = 4$ for days 2, 5, and 7 post-MI). One-way ANOVA with two-tailed Dunnett test was used to determine the statistical significance. **d** Comparison of the expression of Ccr2, 1, 3, 5, and 7 days after MI by FACS. **e** Comparison of the

expression of Trem2 1, 3, 5, and 7 days after MI by FACS. **f** Bars are divided to display the percentage of F4/80+ Ccr2 or Trem2 positive cells in the different experimental days ($n = 4$ for steady-state, days 1 and 7 post-MI; $n = 8$ for day 3 post-MI; $n = 6$ for day 5 post-MI). **g** Quantitative analysis of Trem2+ F4/80+ cells. The sample size is the same as (**f**). One-way ANOVA with two-tailed Dunnett test was used to determine the statistical significance. **$P < 0.01$, ***$P < 0.001$. **h** Comparison of Arg1+ cells within F4/80+Ccr2+ and F4/80+Trem2+ cells, 5 days after MI ($n = 4$ for F4/80+Ccr2+; $n = 8$ for F4/80+Trem2+). Unpaired two-tailed t-test was used to determine the statistical significance. **i** Comparison of mRNA expression levels of pro-inflammatory genes in F4/80+Ccr2+ macrophages at day 1 post-MI ($n = 5$) and F4/80+Trem2+ macrophages at day 5 post-MI ($n = 6$). Unpaired two-tailed t-test was used to determine the statistical significance. *$P < 0.05$, **$P < 0.01$, ***$P < 0.001$.
**j** Comparison of mRNA expression levels of anti-inflammatory genes in F4/80+Ccr2+ macrophages at day 1 post-MI ($n = 5$) and F4/80+Trem2+ macrophages at day 5 post-MI ($n = 6$). Unpaired two-tailed t-test was used to determine the statistical significance. *$P < 0.05$, **$P < 0.01$. All of the bar charts are presented as mean ± SEM. Source data with exact $P$ values for **b**, **c** and **f**–**j** are provided in the Source Data file.

in the infarcted area, while SS-Mφ1 cells were dispersed in the heart at all time points after MI. Considering that tissue-resident macrophages can prevent fibrosis, promote electrical conduction, and facilitate the healing of injured areas[44,45], SS-Mφ1 might contribute to modulating the immune responses in a way different from the other macrophage subsets. Although less prominent, fibroblasts and DCs, which are

known to be involved not only in tissue homeostasis but also in repair and regeneration of the heart[46], were relatively abundant in the infarcted area, suggesting that they may also be involved in infarcted tissue repair.

In conclusion, we present a comprehensive single-cell and spatial transcriptomic landscape of infarcted hearts following MI. We explore

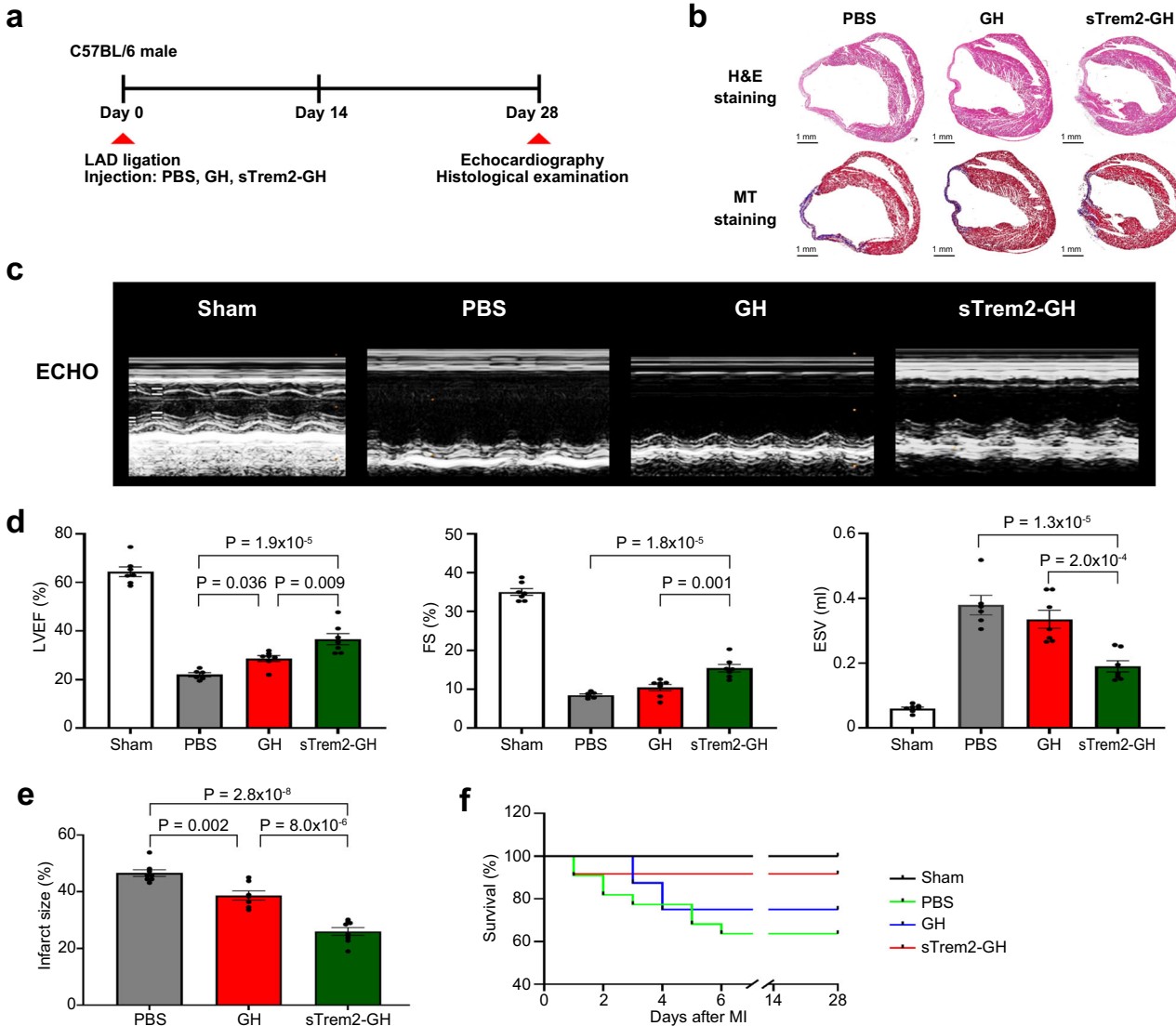

**Fig. 6 | Effect of the injection of soluble Trem2 (sTrem2) on the heart of MI mouse. a** Treatment schedule. We made an injectable gelatin hydrogel (GH) mixed with sTrem2 and then injected it at two sites of the peri-infarcted area after MI. PBS and GH alone were also administered to the control groups. Twenty-eight days post-MI, echocardiography, and histological examination were performed to evaluate the effects of sTrem2. **b** Hematoxylin and eosin (H&E) stained images (upper panel) and Masson's trichrome (MT) stained images (lower panel) of heart tissues of PBS-, GH-, and sTrem2-GH-treated groups at day 28 post-MI. Severe LV dilation was prominent in the control PBS-treated group, while a significant attenuation of LV remodeling with less infarcted wall thinning was observed in the sTrem2-GH-treated group. Results are representative of three different samples. **c** M-mode echocardiography images of sham operation ($n = 7$), PBS- ($n = 6$), GH-

($n = 7$), and sTrem2-GH-treated ($n = 7$) groups at day 28 post-MI. Results are representative of four different samples. **d** Each bar plot represents the left ventricular ejection fraction (left), fractional shortening (middle), and end-systolic volume (right) of the different groups. One-way ANOVA with two-tailed Dunnett test was used to determine the statistical significance. **e** Quantification of the infarct size on day 28 post-MI in PBS- ($n = 8$), GH- ($n = 7$), and sTrme2-GH- ($n = 8$) treated mice. One-way ANOVA with two-tailed Dunnett test was used to determine the statistical significance. **f** Kaplan–Meier survival analysis after MI in sham-operated ($n = 5$), PBS- ($n = 11$), GH- ($n = 11$), sTrem2-GH- ($n = 12$) treated mice. All of the bar charts are presented as mean ± SEM. Source data for **d**–**f** are provided in the Source Data file.

the spatiotemporal dynamics of macrophage heterogeneity and the potential function of Trem2[hi] macrophages in cardiac tissue repair. Our data make important contributions to the current understanding of MI-driven immune responses, and further investigations and discovering the regulatory factors of the Trem2 signaling pathway will help establish novel therapeutic strategies for post-MI HF.

## Methods
### Ethical considerations
All animal experiment procedures were conducted in compliance with the approval of the Institutional Animal Care and Use Committee (IACUC) at the Catholic University School of Medicine (CUMC-2018-0035-07).

### Animal models
We used male wild-type 7–8 week-old C57BL/6 mice purchased from Orient Bio (Gyeonggido, Korea). Mice were housed in a specific pathogen-free facility maintained on a 12 h light-dark cycle at 20–26 °C and 50 ± 10% humidity. Mice weighting 20–22 g were anesthetized with an intraperitoneal injection of a mixture of Zoletil (30 mg/kg; Zoletil 50, Virbac, France) and xylazine (10 mg/kg; Rompun, Bayer Health-Care, Leverkusen, Germany). Mechanical ventilation was carried out using a Harvard Apparatus ventilator and supplemental oxygen to maintain general anesthesia. Mice were intubated with BD Angiocath plus 22GA and placed in an operating table. After a left-sided thoracotomy, experimental MI was induced by permanent ligating the proximal portion of left anterior descending artery (LAD). The protein

of soluble Trem2 (sTrem2, mouse trem2 aa 12–171 His_N-term) was purchased from LifeSpand BioSciences (LSBio, Seattle, WA, USA). To evaluate the efficacy of sTREM2, mice were injected with PBS, gel, or gel containing 12 µg sTrem2 into the myocardium at the infarct border zone after the LAD ligation. Each treatment was injected at two sites of the peri-infarcted area (10 µL per site), right after MI[47,48]. Echocardiography was performed using an Affinity 50 imaging system (Philips, Florida, USA) 28 days after MI. Mice were initially anesthetized with 5% isoflurane and then anesthetized with 1% isoflurane during the echocardiography procedure to maintain the heart rate. The ejection fraction (EF) and fractional shortening (FS) were calculated from M-mode tracings at the level of the papillary muscles to enable consistent measurement at the same anatomic location in different mice.

## Cell staining and flow cytometry

For flow cytometric analysis, heart tissues were digested with collagenase type II solution (Worthington Biochemical Corporation, USA) at a concentration of 500 unit/ml for 40 min at 37 °C and homogenized in gentleMACS Dissociator (Miltenyi Biotec, USA). After homogenization, the tissues were passed through a 40 µm cell strainer, washed with HBSS Buffer, and resuspended in FACS staining buffer. For FACS analysis, the following antibodies were used: FITC anti-mouse CD45 antibody (1 µg/mL; BD; #553080), BV421 anti-mouse Ccr2 (1 µg/mL; BioLegend; #150605), APC anti-human/mouse Trem2 (1 µg/mL; R&D system; #FAB17291A), and PE-Cy7 anti-human/mouse Arg-1 (1 µg/mL; Invitrogen; #25-3697-82). CD45 positive cells were sorted by flow cytometry sorter (Beckman coulter, USA). After staining each sample, FACS analysis was performed using the FACS Canto II (BD) and Flow-jo software (TreeStar Inc., USA).

## Single-cell RNA sequencing library construction

The single-cell library preparation relied on a commercially available droplet method using the 10× Genomic Chromium System (10× Genomic Inc., San Francisco, CA) and Single Cell 3′ v3 Reagent Kit (10× Genomics Inc.) according to the manufacturer's protocol. In brief, dissociated cells were counted by hemocytometer (ThermoFisher) and 16,000 cells per sample were added to each channel. The cells were then partitioned into gel beads in emulsion (GEMs) in the Chromium instrument, where cell lysis and barcoded reverse transcription of RNA occurred. cDNA was synthesized and amplified for 14 cycles. cDNA clean-up was performed using a SPRIselect Reagent Kit (Beckman Coulter, Brea, CA). 50 ng of the amplified cDNA were used for each sample to construct indexed sequencing libraries. Sequencing libraries were sequenced on an Illumina HiSeq2500 platform. This resulted in an average read depth of 93,445 reads/cell (steady-state: 77,078 reads/cell; day 1 post-MI: 87,068 reads/cell; day 3 post-MI: 143,667 reads/cell; day 5 post-MI: 84,228 reads/cell; day 7 post-MI: 75,185 reads/cell).

## Single-cell RNA sequencing data analysis

The sequenced data were processed into expression matrices with the Cell Ranger Single Cell software suite v3.0.1 by 10× Genomics. Raw base-call files from HiSeq2500 sequencer were demultiplexed into library-specific FASTQ files using Cell Ranger *mkfastq* option. Sequencing reads were mapped to the mm10 version 3.0.0 reference, downloaded from the 10× Genomics (https://support.10xgenomics.com/single-cell-gene-expression/software/downloads/latest). Subsequently, cell barcodes and unique molecular identifiers underwent filtering and correction. Reads associated with the retained barcodes were quantified using Cell Ranger *count* option and used to build a transcript count table.

Bioinformatics processing of the scRNA-sequencing data was performed with the R package Seurat (version 3.2.0)[13]. To exclude low-quality cells in scRNA-sequencing, we filtered cells with an expressed gene count fewer than 2% or greater than 98%. Additionally, cells in which more than 10% of reads corresponded to mitochondrial genes

were removed. Data was log-normalized and highly variable features were identified based on a variance stabilizing transformation (VST) method. All datasets (steady-state, day 1, 3, 5, and 7) were then integrated using the canonical correlation analysis (CCA) method, "FindIntegrationAnchors" and "IntegrateData" functions in Seurat. Principal components analysis (PCA) was performed on the integrated datasets. Based on the top 40 principal components (PCs), graph-based clustering was performed using the shared nearest neighbor (SNN) modularity optimization with resolution set to 1.4, and all cells were classified into 34 clusters. Clustering data was then applied followed by uniform manifold approximation and projection (UMAP) allowing the visualization of identified clusters in UMAP plots[49]. Each cell cluster was annotated for their cell type using the SingleR R package[14] (version 1.2.4) and/or assessment of well-known cell-type specific markers. Isolated clusters of cells that expressed the endothelial markers (*Pecam1* and *Kdr*) or fibroblast markers (*Col1a1* and *Col1a2*) were removed from the further analysis as it was likely due to cellular contamination during FACS. Cell cycle analysis was performed by using the "CellCycleScoring" function in Seurat. Differentially expressed gene (DEG) analysis was used to identify significantly differentially expressed genes within each cluster using the logistic regression test for significance and an average $\log_e$ fold-change. Only genes with a positive average $\log_e$ fold-change value (greater than 0.25) and an adjusted $P$ value lower than 0.05 were kept in the analysis. Raw sequencing data generated for scRNA-sequencing have been deposited in Gene Expression Omnibus (GEO) under accession number GSE163129.

Pseudotime trajectories of differentiation were generated using the Monocle3 R package[50] (version 0.2.3). Pseudotime analysis proceeds on the basis that cells undergo biological processes in an asynchronous manner, and thus that cells can be ordered along a calculated trajectory to infer the transcriptional changes throughout the process. Ly6c2hi monocyte was set as a root of the trajectory, and genes that were most differentially expressed in identified clusters were used to assign pseudotime values to individual cells.

## Spatial transcriptome sequencing (ST-seq)

Frozen samples from day 1, 3, 5, and 7 post-MI were embedded in OCT (TissueTek), and cryosectioned into 10 µm slices at −10°C. Sections were placed on chilled Visium Spatial Gene Expression slide (10× Genomics), and adhered by warming the back of the slide. Tissue sections were then fixed in chilled methanol and stained according to the manufactures guideline. Tissues were permeabilized for 30 (Day 1), 6 (day 3), 18 (Day 5), and 12 min (Day 7), which was selected as the optimal time based on Visium Tissue Optimization experiments. For tissue optimization experiments, fluorescent images were taken with a TRITC filter using a confocal microscopy LSM800 (Carl Zeiss, Oberkochen, Germany). Brightfield histology images were taken using a 10× objective on an Olympus BX51 microscope with OlyVia software (Olympus, Tokyo, Japan). cDNA libraries were generated using Visium Spatial Gene Expression slide & Reagent Kit according to the manufacturer's instruction (10X Genomics), and sequenced on a NovaSeq 6000 system (Illumina). This resulted in an average read depth of 90,805 reads/spot (day 1 post-MI: 88,083 reads/ spot; day 3 post-MI: 94,479 reads/spot; day 5 post-MI: 105,089 reads/ spot; day 7 post-MI: 75,568 reads/ spot).

Low FASTQ files and Hematoxylin and Eosin (H&E) stained images were processed using Space Ranger software v1.0.0 (10X Genomic). For mapping of the sequencing data, the mm10 version 3.0.0 *mus musculus* reference genome was used. Raw sequencing data generated for ST-seq have been deposited in Gene Expression Omnibus (GEO) under accession number GSE165857. Bioinformatics processing of the ST-seq data was performed with the R package Seurat (version 3.2.0)[13]. In brief, normalization ("SCTransform" function), dimensionality reduction ("RunPCA" function), graph-based clustering ("FindNeighbors" and

"FindClusters" functions with 15 PCs), UMAP visualization and DEGs analysis was conducted with default parameters. To infer the cell-type composition of each spot, we applied SPOTlight (version 0.1.7)[16] with default parameters. In brief, clusters of the scRNA-seq data were used to train the SPOTlight. Gene set was the union between the marker genes of the cell types along with the top 3,000 variable genes. All markers were used to initialize the model basis, and unit variance normalization was carried out. Cell types contributing <3% to the spot's predicted composition were considered fitting noise and were set as 0.

Signature scores were calculated by taking the mean of the scaled and centered expression value across multiple signature genes using the "AddModuleScore" function in Seurat. Cell type signature scores were generated using the following marker genes for each cell type: cardiomyocytes (*Myh7, Myh6, Actn2, Nkx2-5, Tnni3, Tnnt2*), endothelial (*Cdh5, Ly6c1,* and *Kdr*) and fibroblast (*Col1a1, Pdgfra,* and *Lamc1*).

## Histological and immunohistochemistry analysis

The excised heart tissues were fixed in 4% paraformaldehyde and embedded in paraffin. Tissue blocks were cut into 4 μm serial sections for histological analysis. The sections were stained with hematoxylin and eosin (H&E) and Masson's trichrome (MT) according to the manufacturer's protocol, and myocardial fibrosis and the infarct size were determined by using a digital pathology scanner (Aperio AT Turbo, Leica Biosystems, Germany). For immune-histochemical analysis, after blocking in 10% normal horse serum (Vector Laboratories, USA), the sections were incubated overnight at 4 °C with anti-TREM2 (1:200; Abcam, UK; #ab175525) diluted in PBS. The DAB was used to visualize according to the manufacturer's specifications (DAB chromogen, GBI Labs, USA).

## Co-localization assay

To confirm Trem2 and macrophage Co-localization, the sections performed heat-induced antigen retrieval with Tris/ EDTA buffer pH 9.0 (Abcam). After blocking in 10% normal horse serum (Vector Laboratories, USA), the sections were incubated overnight at 4 °C with anti-CD68 (Abcam; #ab53444) and anti-TREM2 antibody diluted in PBS (1:200). On the second day, secondary antibodies, Alexa 488 anti-rat antibody (1:500; Invitrogen; #A11006) or Alexa 594 anti-Goat antibody (1:500; Invitrogen; #A32758), were treated for 1 h at room temperature. Then, DAPI staining were conducted. Mounting solution used the Aqua-mount solution (DAKO). The stained sections were examined under a confocal microscope (LSM700, ZEISS, Germany).

## Western blotting assay

Total protein was extracted from mouse infarct tissue in lysis buffer (RIPA buffer containing a proteinase inhibitor cocktail). Fifty micrograms of total protein were electrophoretically separated on a 12% sodium dodecyl sulfate-polyacrylamide gel and transferred to a polyvinylidene difluoride membrane using a Trans-Blot Turbo transfer system (Bio-Rad). The membrane was blocked in TBST buffer (10 mM Tris-HCl (pH 8.0), 150 mM NaCl, and 0.02% Tween 20) containing 5% skim milk and then incubated with anti-TREM2 and anti-alpha tubulin (Abcam; #ab125267) antibodies, diluted 1:1000 in 5% skim milk, respectively. The membrane was then incubated with horseradish peroxidase-conjugated anti-mouse (1:5000, Cell Signaling Technology; #7076S) and anti-rabbit secondary antibodies (1:5000; Cell Signaling Technology; #7074S). Specific protein bands were visualized with an enhanced chemiluminescence system (Amersham, Heights, IL, USA) and quantified using ImageJ software.

## Expression of mRNA

Total RNA from Ccr2+ or Trem2+ sorted cells was extracted using RNeasy mini kit (Qiagen, Valencia, USA) according to the manufacturer's protocol. Total RNA was reverse transcribed using reverse transcriptase (Roche, USA). cDNA samples were subjected to real-time quantitative RT-PCR (qRT-PCR) analyses with specific primers for *Il1b, Il6, Il10, Alox15, Cx3cr1, Cxcr3, Cxcr7, Ccl1, Ccl2, Tgfb1,* and *Spp1* (Supplementary Table 3) using a Bio-Rad CFX96 Real-Time PCR detection system (BIO-RAD Laboratories, Inc., Hercules, CA, USA). The Gapdh was amplified as an internal control. Expression level was calculated by ΔΔCt method, and fold changes were obtained using the formula $2^{-\Delta\Delta Ct}$. All samples were run in triplicate.

## Reporting summary

Further information on research design is available in the Nature Research Reporting Summary linked to this article.

## Data availability

The scRNA-seq and ST-seq data sets generated in this study have been deposited in the Gene Expression Omnibus under accession numbers GSE163129 and GSE165857, respectively. Sequencing reads were mapped to the mm10 version 3.0.0 reference, downloaded from the 10x Genomics (https://support.10xgenomics.com/single-cell-gene-expression/software/downloads/latest). Source data are provided with this paper.

## Code availability

All codes used in this manuscript are based on 10X Genomics and public library packages that are listed in the "Methods" section. Relevant codes used for data analysis are available from https://github.com/Junglab-CMC/Macrophage-heterogeneity-after-MI.

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

## Acknowledgements

This study was supported by a grant from the National Research Foundation of Korea [grant numbers 2019R1A5A2027588 (Y.J.C.), 2019R1C1C1004909 (S.H.J.), 2020R1A4A3079570 (K.C.), 2017R1E1A1A01074913 (Y.J.C.), and 2019R1A2C2085516 (K.C.)].

## Author contributions

S.H.J., B.H.H., S.S., K.C., and Y.J.C. wrote the manuscript. K.C. and Y.J.C. conceived and designed the study. B.H.H., E.H.P., S.H.P., C.W.K., E.K., E.C., and I.J.C. participated in the acquisition of data, sample preparation, and clinical reviews. S.H.J. and S.S. carried out molecular genetic studies. S.H.J. and S.S. carried out the bioinformatics analysis. F.K.S. provided intellectual input and discussed results and strategy. All authors read and approved the final manuscript.

## Competing interests

The authors declare no competing interests.
