## [Peer Review File · Nature Communications]

REVIEWER COMMENTS

Reviewer #1 (Remarks to the Author):

In this manuscript, authors have analyzed CD45+ cells after MI using scRNA-seq and Visium. Previous studies have shown that macrophage populations and functions are temporally changed after MI. Therefore, it is important to further characterize the functional and phenotypic diversities in cardiac macrophages after MI. The authors show the temporal changes in macrophage subsets and provide data suggesting TREM2 may have a cardioprotective function after MI.

Major points:

In Figure 2, the authors used the Visium system to analyze the spatial transcriptome of the myocardium. The authors described, "Most of the mouse heart cell were cardiomyocytes...". This is misleading and contradicts the previous careful studies of the cardiac cell populations (AJP 293:H1883-H1891, 2007; Circ Res 118:400-409, 2016). Nonmyocytes, including fibroblasts and macrophages, are known to be scattered in the myocardium (Cell 183:94-109.e123, 2020.). Because the resolution of methods is noted as 1-10 cells, it is likely the inclusion of RNA from cardiomyocytes, which would cluster myocyte-containing spots as a cluster expressing myocyte markers, resulting in an overestimation of the number of myocytes. Authors should consider revising the part to more precisely state what can be concluded from the results.

The time course of immunostaining of Tmem2 and that of western do not match well. Does this reflect the changes in TREM2+ macrophage number rather than expression levels of TREM2 in macrophages? Also, on single-cell data TREM2+ macrophage fraction may stay less than 50% even on days 3 and 5, though the FACS showed otherwise. Did the sections shown contain only these TREM2+ macrophage populations?

While the data shown in Fig. 6 suggest an overall beneficial effect of sTREM2, it remains unknown as to how TREM2 improved the consequence of MI. What are the cells that are affected by sTREM2?

M1/M2 dichotomy was proposed based on the findings of cultured macrophages, and it is now clear that the dichotomy cannot be used to classify macrophages in tissues in vivo (Circulation Research. 2016;119:414-417). The authors label tissue-resident macrophages as M2 and suggest that monocyte-derived macrophages differentiate into M2 after MI. This is problematic for several reasons. First, the marker gene expression pattern of cardiac tissue resident macrophages may not fit the M2 markers. For instance, the authors showed a typical M2 marker, Arg1, was high in TREM2+ macrophages, which may be a transient population that appears late (i.e., 3-7 days) after MI. They are likely to be monocyte-derived cells as opposed to the fetal liver-derived tissue resident macrophages in healthy heart. As such, it is unclear why tissue resident macrophages in the steady-state and the macrophages appear late and resembles tissue-resident macrophages should be considered M2. Second, the differentiation trajectories inferred from scRNA data do not necessarily show the actual differentiation course of cells. Because the cells are clustered based on the similarities in transcriptomes and projected on UMAP, and the trajectories are mapped on it, the results may look like a sequential differentiation from Ly6Chi to resident macrophages. However, actual differentiation processes need further biological and computational analyses. For instance, the late macrophage populations may predominate on days 5-7, and they may have transcriptomes similar to those of the resident macrophages. However, they might disappear after this time point and may never differentiate into the resident macrophages. It is also possible that monocytes recruited on day 1 become M1, but those recruited on day 7 directly become TREM2+ cells. As such, the results shown in this study are not sufficient to support the continuum model of differentiation from M1 to M2.

Minor

p. 11 l.244 Fig.5b should be 5c.

Reviewer #2 (Remarks to the Author):

The authors investigated the spatiotemporal dynamics of MI-associated immune cells in a murine model of MI using single-cell RNA-sequencing and spatial transcriptomics. They describe the heterogeneity, compositional changes, and approximate spatial localization of immune cell populations in their model. They identify upregulation of Trem2 expression in macrophages and show that in vivo injection of soluble Trem2 resulted in functional and structural improvement in infarcted hearts. These results are of significance to the field.

In general, the work supports the conclusions and claims and there are no flaws in the data analysis, interpretation and conclusions that prohibit publication or require revision. The authors used standard scRNA-seq computational analysis pipelines and methodology meets the expected standards in this field. There is sufficient detail provided in the methods for the work to be reproduced.

I have a few minor questions/suggestions for the authors:

Line 117: "Three macrophage sub-clusters (clusters 1, 3, and 13) were predominant in the steady-state (77.9% of the total monocyte/macrophages), suggesting that they are tissue-resident macrophages (termed Res-M ϕ clusters, Fig. 3b)." Not sure that the predominance of these subclusters proves that they are tissue-resident macrophages. Are there any markers or techniques (eg macrophage-marker IV negative labeling) that would prove this?

Macrophage clustering seen in ST-seq is interesting, but of course it should be stated that this is not technically single-cell, so the authors are using cell type signatures as an approximate approach to identify cell types. Nothing to do about this, just that this caveat should be stated in the manuscript.

Line 179: "When Ly6c2hi monocytes were set as the root of the trajectory" What is the justification for setting Ly6c hi monocytes as the root?

Line 192: "Importantly, when we checked the expression of candidate genes in injured heart tissues by ST-seq, marker genes for each subset were enriched in the infarcted area of the heart in a time-dependent manner, while Res-M ϕ 1 cells were dispersed in the heart for all time points after MI (Fig. 3e, Supplementary Fig. 5). Supp Fig 5 uses four putative marker genes for res-Mac1 "subset", but the expression for each gene is rather sparse and does not prove that these genes are expressed specifically by macrophages.

Imputed trajectory analyses (especially using the original Monocle algorithm) are not without caveats (eg artificially forcing trajectories), which should be stated somewhere in the paper.

Do the authors envision that injection of soluble Trem2 would have translational relevance? What cells other than macrophages might respond to Trem2?

Reviewer #3 (Remarks to the Author):

Single-cell transcriptomics unveils the spatiotemporal dynamics of macrophage heterogeneity and a potential role of Trem2hi macrophages in infarcted hearts
In this manuscript, the authors have applied the single-cell RNA sequencing and spatial transcriptome sequencing to identify different populations of immune cells that operate at different time points and with different dynamics following MI. In particular,

macrophages have emerged as the most dominant and enriched population able to release Trem2 in the later stages post-MI. By in vivo injection of soluble Trem2, the authors show functional and structural improvements of the infarcted heart.

This paper provides an excellent overview of the interplay of different types and subtypes of immune cells following MI by using innovative techniques. Overall, the manuscript is fluid to read and very informative. Nonetheless, there are a few concerns that should be addressed.

Major Remarks

1. Fig.1d, authors present proportion of macrophages during the 7 days following MI. However, expansion of infarct size associated with MI time frame has not been considered here. Therefore, addition of table with information of infarct size, cell population (cell counts and percentage).
2. Fig.2a and 2b, representative image of day 5 post-MI sample is not appropriate.
3. Fig.2b, H&E image is not enough to encompass MI related injury. For better interpretation, masson's trichrome and TTC staining are recommended. Moreover, the authors explain in the manuscript, pag.7, that day1 post-MI, there is an increase in monocyte recruitment, why the monocyte-specific score is not mentioned in the fig either in the supplementary material?
4. In Fig.3B, the authors show a higher presence of resident macrophages at a steady-state, a decrease after MI and restored at days 5-7. Authors need to describe whether the resident macrophages are restored due to proliferation of the remaining resident cells or replacement.
5. In Fig.5f, expression of CCR2 and Trem2 are inversely correlated, dose CCR2 positive cells are gradually transit to Trem2 macrophages? Or is Trem2 are fresh population at later stage of MI? Moreover, the late macrophages increase at days 5 and 7, don't express CCR but they are not mentioned as resident macrophages. What is their origin?
6. Authors provide IHC image and co-staining of Trem2 and cardiac macrophages, however, whether macrophages are only cell source of Trem2 is not clear. Measurement of Trem2 expression in all cardiac cell fraction is recommended.
7. Please add the quantification of WB in Fig.5b. Additionally, it is required a better description of the full length and soluble form of Trem2.
8. Authors showed soluble fraction of Trem2 increased in later stage of MI. What about in secretome or plasma, is Trem2 commonly detected in animal with MI or heart failure patient?
9. In Fig.5c, it would be interesting to integrate the staining for cardiomyocytes. Considering the starting demonstrations in which late macrophages co-localize with the damaged area (Fig.3E), a staining to highlight the infarcted area should be performed in combination with CD68 and Trem2.
10. Fig.6 should be integrated with additional information such as ratio LV mass/BW and a table with morphometric and echocardiographic characteristics of the mice (mortality rate, infarct size, body weight, IVSd, IVSs, LVIDd, LVIDs; etc.). Table would be great addition.
11. Does sTrem2 with GH specifically targeting macrophages? A distribution of the Trem2 with GH in myocardium should be presented. Moreover, how authors select dose of Trem2 need to be better described.
12. The expression of sTrem2 is increased in late stage (7days) of MI, what about in

chronic stage such as 28 days. Authors inject sTrem2 right after MI and it showed protective effect, what is mechanism behind? Is it due to transition of macrophages phenotype or anti-inflammatory response to entire heart? Authors need more evidence to clarify the cardioprotective effect of Trem2.

13. Tremhi macrophages exhibit more M2 phenotypes with increased Arg1 level, however this result is not sufficient to conclude Tremhi macrophages are anti-inflammatory macrophages. Facs analysis of Tremhi macrophages are required for better understanding of the effect of Trem on macrophage polarization.

14. Despite M2 macrophages present anti-inflammatory characters, they often show also pro-fibrotic features. Authors didn't show any results related to fibrosis or fibroblasts. Additional data about the effect of sTrem2 on fibrosis are required.

15. About Fig.6C, it would be interesting to show the results compared to the control group of mice without LAD ligation. Moreover, it seems that the gelatin hydrogel without Trem2 significantly improves the LVEF and FS compared to the PBS treatment. Authors need clarify that in discussion.

Minor Remarks

The authors should describe the number of biological replicates for each experiment. In the introduction is mentioned the nomenclature M1 and M2 that is obsolete. The in vivo environment is more complex, as demonstrated by this paper, so this nomenclature can be limited only to an in vitro environment. I would suggest to the authors to specify this or to edit it (PMID: 27458196).

Since Trem2 is the crucial molecule discussed in this manuscript, this should be mentioned and briefly described also in the introductory part, not only in the discussions.

The results of Fig.1 and Fig.2 should be better discussed. For Fig.2 the conclusion is completely missed.

Page 11, line 244, there is a mistake in the mentioned Figure. It should be 5c and not 5b.

Fig.5d-f requires an extensive explanation.

In material and methods, paragraph co-localization assay, the reference of the secondary antibodies is missing.

Response to reviewer comments

Reviewer #1

In this manuscript, authors have analyzed CD45⁺ cells after MI using scRNA-seq and Visium. Previous studies have shown that macrophage populations and functions are temporally changed after MI. Therefore, it is important to further characterize the functional and phenotypic diversities in cardiac macrophages after MI. The authors show the temporal changes in macrophage subsets and provide data suggesting TREM2 may have a cardioprotective function after MI.

Comment 1. In Figure 2, the authors used the Visium system to analyze the spatial transcriptome of the myocardium. The authors described, “Most of the mouse heart cell were cardiomyocytes...”. This is misleading and contradicts the previous careful studies of the cardiac cell populations (AJP 293:H1883-H1891, 2007; Circ Res 118:400-409, 2016). Nonmyocytes, including fibroblasts and macrophages, are known to be scattered in the myocardium (Cell 183:94-109.e123, 2020.). Because the resolution of methods is noted as 1-10 cells, it is likely the inclusion of RNA from cardiomyocytes, which would cluster myocyte-containing spots as a cluster expressing myocyte markers, resulting in an overestimation of the number of myocytes. Authors should consider revising the part to more precisely state what can be concluded from the results.

Response: We agree with the reviewer’s comment on the ST-seq data. As the reviewer suggested, we have reanalyzed the ST-seq data using SPOTlight, a nonnegative matrix factorization (NMF)-based spatial deconvolution framework (PMID: 33544846), to infer the cell-type composition of each spot. SPOTlight identified cell-type-specific gene expression signatures from scRNA-seq data, which were subsequently used to deconvolute ST-seq spots. Based on the SPOTlight analysis, we have revised Figure 2 and added Supplementary Figures 2-5 and 11. Because our scRNA-seq data included only cytometry-sorted CD45⁺ leukocytes,

statements regarding the cardiomyocyte have been removed from the manuscript. In addition, we have replaced the ST-seq data of the day 5 post-MI sample, which was noted as an inappropriate image by reviewer #3.

The corrections in the Result section of the revised manuscript are as follows. “We then applied SPOTlight (PMID: 33544846), a nonnegative matrix factorization (NMF)-based spatial deconvolution framework, to infer the cell-type composition of each spot (Fig. 2b). Notably, neutrophils highly infiltrated into the infarcted area on day 1 (41.2% on average) but their numbers were then decreased rapidly (18.7, 10.6, and 9.5% on days 3, 5, and 7, respectively), which was consistent with the scRNA-seq data (Fig. 2b and 2c). On the contrary, macrophages were dispersed across the whole heart rather than clustered in the infarcted area on day 1 (Fig. 2b). However, from day 3, macrophages infiltrated into the infarcted area, and their abundance peaked at late MI (days 5 and 7; 18.8% on average) (Fig. 2c). Monocytes and fibroblasts also infiltrated the infarcted area of the heart in a time-dependent manner (Supplementary Fig. 1-5). Other immune cell populations, such as B-cell and T-cell populations, were always dispersed across the entire mouse heart (not clustered in the infarcted area), with a very low abundance (Supplementary Fig. 2-5). Together with the scRNA-seq showing immune cell temporal dynamics after MI, ST-seq analysis with deconvolution algorithm further provided their spatial heterogeneity and spatiotemporal dynamics; our ST-seq data clearly showed that monocytes and neutrophils infiltrated into the infarcted area at early MI while macrophages and fibroblasts acted oppositely.” (page 7-8, line 119-135)

We have also added a statement on the caveat for the ST-seq in the Discussion section as follows. “Although ST-seq used in this study does not provide spatial expression profiles at true single-cell resolution (1–10 cell resolution on average per spot), ST-seq analysis with deconvolution algorithm revealed that the proportions of monocytes, neutrophils, and macrophages increased or decreased in a time-dependent manner, which was consistent with

previous reports.” (page 20, line 422-426)

Comment 2. The time course of immunostaining of Trem2 and that of western do not match well. Does this reflect the changes in TREM2⁺ macrophage number rather than expression levels of TREM2 in macrophages? Also, on single-cell data TMEM2⁺ macrophage fraction may stay less than 50% even on days 3 and 5, though the FACS showed otherwise. Did the sections shown contain only these TMEM2⁺ macrophage populations?

Response: We appreciate this valuable comment. In Figure 5b, we intended to confirm the expression dynamics of Trem2 over time after MI, in which we used the anti-Trem2 antibody targeting common C-terminals contained in both full-length Trem2 and its soluble form (sTrem2). The results showed that the expression of full-length Trem2 peaked on day 5 post-MI and decreased thereafter, whereas that of sTrem2 appeared on day 3 and peaked on day 7. Thus, the expression of total Trem2 (full-length Trem2 and sTrem2) steadily increased, peaking 5 days post-MI and decreasing thereafter. We believe these results are compatible with those obtained in Trem2 IHC analysis and co-localization assay. To clarify this, we have changed “upper bands from Trem2” to “full-length Trem2” and added the quantitative analysis data for comparison of relative Trem2 levels in the revised Figure 5b and 5c.

<Revised Figure 5a-5c>

The discrepancy in Trem2^{hi} macrophage populations between scRNA-seq and FACS data

seems to be due to the difference between the two techniques. In scRNA-seq, as the reviewer mentioned in comment 4, CD45⁺ leukocytes isolated from infarct and peri-infarct areas were clustered based on the similarities in transcriptomes, not Trem2 expression. Thus, the gene signature-based clustering can cause an underestimation of the number of Trem2-expressing macrophages. Indeed, some macrophages expressing Trem2 were grouped into other macrophage subsets, such as Transient-Mφ2, Transient-Mφ3, and Late-Mφ1 (Revised Supplementary Figure 7). In contrast, in FACS analysis, single-cell suspensions were sorted into CD45⁺F4/80⁺ macrophages to collect Trem2^{hi} macrophage subsets specifically. Although a relatively high Trem2^{hi} macrophage proportion was observed in FACS analysis compared to scRNA-Seq, the trends (dominance in late phase post-MI) were consistent for both methods.

<Revised supplementary figure 7>

Comment 3. While the data shown in Fig. 6 suggest an overall beneficial effect of sTREM2, it remains unknown as to how TREM2 improved the consequence of MI. What are the cells

that are affected by sTREM2?

Response: We appreciate the reviewer's valuable comment. Regarding this issue, we first analyzed the expression level of Arginase 1 (Arg1) to characterize the polarizing status of the Ccr2^{hi} and Trem2^{hi} macrophage subsets. Significantly higher expression levels of Arg1 were observed in Trem2^{hi} macrophages than that in Ccr2^{hi} macrophages (revised Fig. 5h). Next, we quantified inflammatory and anti-inflammatory mRNA expression levels in Ccr2^{hi} and Trem2^{hi} macrophages (revised Fig. 5i and 5j). Ccr2^{hi} macrophages exhibited higher expression levels of pro-inflammatory cytokines, such as *IL-1 β* and *IL-6*, whereas Trem2^{hi} macrophages expressed higher levels of anti-inflammatory cytokines, such as *IL-4*, *IL-10*, and *Alox15*. In addition, Trem2^{hi} macrophages augmented anti-inflammatory chemokines, such as *Ccl1* and *Cxcr1*, as well as anti-inflammatory signature genes such as *TGF- β* . By contrast, Ccr2^{hi} macrophages expressed higher levels of pro-inflammatory chemokines, such as *Ccl2*, *Cxcr3*, and *Cxcr7*. These results indicate the anti-inflammatory nature of Trem2^{hi} macrophages in contrast with early-stage Ccr2^{hi} macrophages. Interestingly, Trem2^{hi} macrophages exhibited a higher expression level of osteopontin (*Spp1*), which is related to pro-fibrotic potential in regulating post-MI LV remodeling (PMID: 11375279, 31411676). These results indicate that Trem2^{hi} macrophages may have anti-inflammatory macrophage characteristics.

<Revised figure 5i and 5j>

Although we found the beneficial anti-remodeling effect of sTrem2 post-MI, it will be a difficult task to investigate the underlying mechanisms, such as what type of cells are affected by sTrem2 and how sTrem2 induces intracellular signaling. Late-stage Trem2 macrophages

may secrete sTrem2 and can affect the macrophage itself via autocrine regulation. To investigate how sTrem2 affects the macrophage, we tested the polarizing effect of sTrem2 on thioglycolate-elicited peritoneal macrophages from C57BL/6 mice. When peritoneal macrophages were activated toward a pro-inflammatory phenotype by lipopolysaccharide (LPS, 1 $\mu\text{g}/\text{mL}$) and interferon- γ (IFN- γ , 4 ng/mL) treatments, *IL-12B* and *iNOS* (pro-inflammatory markers) were upregulated while *Arg1* (anti-inflammatory marker) was downregulated. On the contrary, sTrem2 (200 ng/ml) treatment after LPS and IFN- γ treatments decreased or increased the expression levels of pro-inflammatory and anti-inflammatory markers, respectively (see the figure below). This finding suggests that sTrem2 affects macrophage polarization, but since further detailed research should be performed to draw a firm conclusion, we did not include these *in vitro* data in the revised manuscript. We agree with the reviewer's concern about the Trem2 function in MI, but the exploration of more possibilities is beyond our capacity at this time and will be carried out in future studies. This statement has been added to the Discussion section of the revised manuscript as follows: "Further studies are, however, needed to systematically evaluate the Trem2^{hi} macrophage origin, the cells affected by sTrem2, and the related intracellular signaling pathways, to enable a better understanding of its regenerative effect in infarcted hearts, before conducting human clinical trials." (page 19, line 397-400)

< *in vitro* polarization in peritoneal macrophages >

Comment 4. M1/M2 dichotomy was proposed based on the findings of cultured macrophages, and it is now clear that the dichotomy cannot be used to classify macrophages in tissues *in vivo* (Circulation Research. 2016;119:414–417). The authors label tissue-resident macrophages as M2 and suggest that monocyte-derived macrophages differentiate into M2 after MI. This is problematic for several reasons. First, the marker gene expression pattern of cardiac tissue resident macrophages may not fit the M2 markers. For instance, the authors showed a typical M2 marker, Arg1, was high in TREM2⁺ macrophages, which may be a transient population that appears late (i.e., 3-7 days) after MI. They are likely to be monocyte-derived cells as opposed to the fetal liver-derived tissue resident macrophages in healthy heart. As such, it is unclear why tissue resident macrophages in the steady-state and the macrophages appear late and resembles tissue-resident macrophages should be considered M2. Second, the differentiation trajectories inferred from scRNA data do not necessarily show the actual differentiation course of cells. Because the cells are clustered based on the similarities in transcriptomes and projected on UMAP, and the trajectories are mapped on it, the results may look like a sequential differentiation from Ly6Chi to resident macrophages. However, actual differentiation processes need further biological and computational analyses. For instance, the late macrophage populations may predominate on days 5-7, and they may have transcriptomes similar to those of the resident macrophages. However, they might disappear after this time point and may never differentiate into the resident macrophages. It is also possible that monocytes recruited on day 1 become M1, but those recruited on day 7 directly become TREM2⁺ cells. As such, the results shown in this study are not sufficient to support the continuum model of differentiation from M1 to M2.

Response: We agree with the reviewer's concern. As suggested, the M1/M2 dichotomy is no longer used in the revised manuscript and related sentences were consistently changed in the Introduction as follows: "However, this dichotomous classification was obsolete, as *in vivo*

environment of macrophages are more complex (PMID: 27458196). Therefore, a more precise evaluation is needed for accurate characterization of the dynamics of macrophage heterogeneity during the acute period of MI.” (page 4, line 61-64)

The following was added to the Discussion: “third, the single-cell trajectory analysis supports a sequential differentiation from $Ly6c2^{hi}$ monocytes to Late-M ϕ rather than the obsolete dichotomous M1-M2 paradigm” (page 17, line 335-337) Filip K. Swirski, the author of the paper the reviewer mentioned and one of the authors of this manuscript, has read and approved this manuscript.

In addition, the trajectory analysis was re-performed by excluding the tissue-resident macrophages (see the figure below), and caveats for inferred trajectories have been added in the Discussion as follows. “Although our inferred trajectories harbored a limitation that $Ly6c2^{hi}$ monocytes were artificially designated into root state with simple tree-structure models (PMID: 30936559), the pseudo-time analysis of monocytes/macrophages matched the order of sampling time-points after MI, supporting a sequential differentiation from $Ly6c2^{hi}$ monocytes to Late-M ϕ rather than the conventional dichotomous M1-M2 paradigm.” (page 18, line 366-370)

<Revised figure 3c and 3d>

Comment 5. (Minor points) p. 11 1.244 Fig.5b should be 5c.

Response: We have corrected this mistake in the revised manuscript. Thank you.

Reviewer #2

The authors investigated the spatiotemporal dynamics of MI-associated immune cells in a murine model of MI using single-cell RNA-sequencing and spatial transcriptomics. They describe the heterogeneity, compositional changes, and approximate spatial localization of immune cell populations in their model. They identify upregulation of Trem2 expression in macrophages and show that in vivo injection of soluble Trem2 resulted in functional and structural improvement in infarcted hearts. These results are of significance to the field. In general, the work supports the conclusions and claims and there are no flaws in the data analysis, interpretation and conclusions that prohibit publication or require revision. The authors used standard scRNA-seq computational analysis pipelines and methodology meets the expected standards in this field. There is sufficient detail provided in the methods for the work to be reproduced.

Response: We thank the reviewer for recognizing this work as worthy of publication.

Comment 1. (Minor points) Line 117: “Three macrophage sub-clusters (clusters 1, 3, and 13) were predominant in the steady-state (77.9% of the total monocyte/macrophages), suggesting that they are tissue-resident macrophages (termed Res-M ϕ clusters, Fig. 3b).” Not sure that the predominance of these subclusters proves that they are tissue-resident macrophages. Are there any markers or techniques (eg macrophage-marker IV negative labeling) that would prove this? Macrophage clustering seen in ST-seq is interesting, but of course it should be stated that this is not technically single-cell, so the authors are using cell type signatures as an approximate approach to identify cell types. Nothing to do about this, just that this caveat should be stated in the manuscript.

Response: We appreciate this comment. In a recent work by Dick *et al.* (PMID: 30538339), scRNA-seq analysis of mouse heart tissue at steady state identified four distinct macrophage

clusters (TIMD4, MHC-II, ISG, and CCR2 M ϕ subsets) as tissue-resident cardiac macrophages. The fate-mapping analysis further showed that CCR2⁻TIMD4⁺ M ϕ population was maintained independently of monocytes: CCR2⁻TIMD4⁺MHCII^{hi} M ϕ were partially replaced by monocytes and CCR2⁺MHCII^{hi} M ϕ were fully replaced by monocytes over time. In our study, two specific findings seemed to support our interpretation: 1) Three macrophage sub-clusters (clusters 1, 3, and 13) were predominant in the steady-state, not post-MI state; 2) cluster 3 macrophages (Res-M ϕ 1) showed elevated expression of TIMD4 M ϕ -related genes (*Lyve1*, *F13a1*, *Cbr2*, *Folr2*, and *Timd4*) and cluster 1 macrophages (Res-M ϕ 2) highly expressed MHC-II M ϕ -related genes (*H2-Eb1*, *H2-Aa*, *H2-Ab1*). To distinguish the monocyte- and embryo-derived resident macrophages more clearly, we have consistently changed the term from tissue-resident macrophages (Res-M ϕ clusters) to steady-state macrophages (SS-M ϕ clusters) in the revised manuscript. We have also added Supplementary Figure 7, and the relevant results were more clearly described as follows: “Expectedly, *Ccr2*, a marker of monocyte-derived macrophages, was not expressed in the SS-M ϕ 1, whereas it was partially expressed in the SS-M ϕ 2 (Supplementary Fig. 7) (PMID: 30538339).” (page 9, line 155-157) Regarding the ST-seq, we have stated the caveat for the ST-seq in the Discussion section as follows. “Although ST-seq used in this study does not provide spatial expression profiles at true single-cell resolution (1–10 cell resolution on average per spot), ST-seq analysis with deconvolution algorithm revealed that the proportions of monocytes, neutrophils, and macrophages increased or decreased in a time-dependent manner, which was consistent with previous reports.” (page 20, line 422-426)

Comment 2. (Minor points) Line 179: “When Ly6c^{2hi} monocytes were set as the root of the trajectory” What is the justification for setting Ly6c^{hi} monocytes as the root?

Response: Mouse Ly6c^{hi} monocytes share several properties with human CD16⁻CD14⁺

monocytes and they are inflammatory. In addition, it is well known that when the Ly6c^{hi} monocytes enter the circulation, contribute to excessive monocytosis, preferentially accumulate around lesions, and differentiate into macrophages after MI (PMID: 23307733, 17200719, and 22144566). Therefore, we set the Ly6c^{2hi} monocytes as the root state of the trajectory analysis. This statement has been added to the Results section of the revised manuscript as follows. “Ly6c^{2hi} monocytes share several properties with human CD16⁻CD14⁺ monocytes, and it is well known that they enter the circulation, contribute to excessive monocytosis, preferentially accumulate in lesions, and differentiate into macrophages after MI.” (page 11, line 207-210)

Comment 3. (Minor points) Line 192: “Importantly, when we checked the expression of candidate genes in injured heart tissues by ST-seq, marker genes for each subset were enriched in the infarcted area of the heart in a time-dependent manner, while Res-Mφ1 cells were dispersed in the heart for all time points after MI (Fig. 3e, Supplementary Fig. 5). Supp Fig 5 uses four putative marker genes for res-Mac1 “subset”, but the expression for each gene is rather sparse and does not prove that these genes are expressed specifically by macrophages. Imputed trajectory analyses (especially using the original Monocle algorithm) are not without caveats (eg artificially forcing trajectories), which should be stated somewhere in the paper.

Response: We appreciate this comment. In this revision, we have reanalyzed the ST-seq data using SPOTlight, a nonnegative matrix factorization (NMF)-based spatial deconvolution framework (PMID: 33544846), to infer the cell-type composition of each spot. SPOTlight identified cell-type-specific gene expression signatures from scRNA-seq data, which were subsequently used to deconvolute ST-seq spots. Based on the SPOTlight analysis, we have revised Figure 2 and added Supplementary Figures 2-5 and 11. In addition, we have replaced the ST-seq data of the day 5 post-MI sample, which was noted as an inappropriate image by

reviewer #3.

When we analyzed the proportion of each macrophage sub-cluster via ST-seq using SPOTlight, proportions for each macrophage subset were enriched in the infarcted area of the heart in a time-dependent manner, while SS(Res)-M ϕ 1 cells were dispersed in the heart for all time points after MI (Revised Supplementary Figure 11). Considering that not only four putative marker genes for SS(Res)-M ϕ 1 but all marker genes from scRNA-seq and 3,000 variable genes from ST-seq were used for SPOTlight, we believe that the revised Supplementary Figure 11 supports our interpretation.

< Revised supplementary figure 11 >

We have also added a statement on the caveats for inferred trajectories to the Discussion section as follows. “Although our inferred trajectories harbored a limitation that Ly6c2^{hi} monocytes were artificially designated into root state with simple tree-structure models (PMID: 30936559),

the pseudo-time analysis of monocytes/macrophages matched the order of sampling time-points after MI, supporting a sequential differentiation from Ly6c^{hi} monocytes to Late-Mφ rather than the conventional dichotomous M1-M2 paradigm.” (page 18, line 366-370)

Comment 4. (Minor points) Do the authors envision that injection of soluble Trem2 would have translational relevance? What cells other than macrophages might respond to Trem2?

Response: Thank you for your inspirational comment. Although we found the beneficial anti-remodeling effect of sTrem2 post-MI, it will be a difficult task to investigate the underlying mechanisms, such as what type of cells are affected by sTrem2 and how sTrem2 induces intracellular signaling. Late-stage Trem2 macrophages may secrete sTrem2 and can affect the macrophage itself via autocrine regulation. To investigate how sTrem2 affects the macrophage, we tested the polarizing effect of sTrem2 on thioglycolate-elicited peritoneal macrophages from C57BL/6 mice. When peritoneal macrophages were activated toward a pro-inflammatory phenotype by lipopolysaccharide (LPS, 1 µg/mL) and interferon-γ (IFN-γ, 4 ng/mL) treatments, *IL-12B* and *iNOS* (pro-inflammatory markers) were upregulated while *Arg1* (anti-inflammatory marker) was downregulated. On the contrary, sTrem2 (200 ng/ml) treatment after LPS and IFN-γ treatments decreased or increased the expression levels of pro-inflammatory and anti-inflammatory markers, respectively (see the figure below). This finding suggests that sTrem2 affects macrophage polarization, but since further detailed research should be performed to draw a firm conclusion, we did not include these in vitro data in the revised manuscript. We agree with the reviewer’s concern about the Trem2 function in MI, but the exploration of more possibilities is beyond our capacity at this time and will be carried out in future studies. This statement has been added to the Discussion section of the revised manuscript as follows: “Further studies are, however, needed to systematically evaluate the Trem2^{hi} macrophage origin, the cells affected by sTrem2, and the related intracellular signaling pathways, to enable

a better understanding of its regenerative effect in infarcted hearts, before conducting human clinical trials.” (page 19, line 397-400)

< *in vitro* polarization in peritoneal macrophages >

Reviewer #3

Single-cell transcriptomics unveils the spatiotemporal dynamics of macrophage heterogeneity and a potential role of Trem2^{hi} macrophages in infarcted hearts. In this manuscript, the authors have applied the single-cell RNA sequencing and spatial transcriptome sequencing to identify different populations of immune cells that operate at different time points and with different dynamics following MI. In particular, macrophages have emerged as the most dominant and enriched population able to release Trem2 in the later stages post-MI. By in vivo injection of soluble Trem2, the authors show functional and structural improvements of the infarcted heart. This paper provides an excellent overview of the interplay of different types and subtypes of immune cells following MI by using innovative techniques. Overall, the manuscript is fluid to read and very informative. Nonetheless, there are a few concerns that should be addressed.

Response: We thank the reviewer for recognizing this work as worthy of publication.

Comment 1. Fig.1d, authors present proportion of macrophages during the 7 days following MI. However, expansion of infarct size associated with MI time frame has not been considered here. Therefore, addition of table with information of infarct size, cell population (cell counts and percentage).

Response: As the reviewer suggested, we have added the additional table containing infarct size and cell population information to the revised manuscript as Supplementary Table 2.

Supplementary Table 2. The portions of cells in each of the 12 broad cell types according to the time-point after MI.

	Infarct size (%)*	Macrophage	Neutrophil	B cell	Monocyte	NK cell	Cd209a+ DC	T cell	Xcr1 DC	Migratory DC	ILC2	Plasma cell	Mast cell	Total
Steady-state	-	3,860	204	1,196	245	568	119	281	37	4	57	3	7	6,581
		58.7%	3.1%	18.2%	3.7%	8.6%	1.8%	4.3%	0.6%	0.1%	0.9%	0.0%	0.1%	100%
Day 1 post-MI	29.91 ± 5.84	1,831	4,009	384	564	87	250	164	48	11	14	0	1	7,363
		24.9%	54.4%	5.2%	7.7%	1.2%	3.4%	2.2%	0.7%	0.1%	0.2%	0.0%	0.0%	100%
Day 3 post-MI	31.65 ± 4.64	2,968	660	99	284	65	272	18	35	42	2	0	1	4,446
		66.8%	14.8%	2.2%	6.4%	1.5%	6.1%	0.4%	0.8%	0.9%	0.0%	0.0%	0.0%	100%
Day 5 post-MI	33.18 ± 1.52	6,110	126	75	259	84	440	23	61	67	4	19	3	7,271
		84.0%	1.7%	1.0%	3.6%	1.2%	6.1%	0.3%	0.8%	0.9%	0.1%	0.3%	0.0%	100%
Day 7 post-MI	33.58 ± 3.08	7,032	136	81	155	177	384	107	87	64	28	47	18	8,316
		84.6%	1.6%	1.0%	1.9%	2.1%	4.6%	1.3%	1.0%	0.8%	0.3%	0.6%	0.2%	100%

*Infarct sizes are shown as the mean ± SD (n = 4).

Comment 2. Fig.2a and 2b, representative image of day 5 post-MI sample is not appropriate.

Response: As the reviewer suggested, we have re-generated the ST-seq data of the day 5 post-MI sample. Accordingly, the related figures (revised figures 2 and 3) and contents (revised supplementary figures 1-5 and 10-11) have been consistently changed in the revised manuscript.

<Revised figure 2a and 2b>

Comment 3. Fig.2b, H&E image is not enough to encompass MI related injury. For better interpretation, masson's trichome and TTC staining are recommended. Moreover, the authors explain in the manuscript, pag.7, that day1 post-MI, there is an increase in monocyte recruitment, why the monocyte-specific score is not mentioned in the fig either in the supplementary material?

Response: As the reviewer suggested, we have added Masson's trichrome staining to the revised Figure 2b (see the figure above). We did not include the TTC staining in the revised manuscript because it is difficult to consistently compare spatial transcriptome profiles of infarcted hearts and it does not significantly support our intentions. For the reviewer's information, we have attached the TTC staining results at different time points below.

Regarding the monocyte-specific score, we have reanalyzed the ST-seq data using SPOTlight, a nonnegative matrix factorization (NMF)-based spatial deconvolution framework (PMID: 33544846), to infer the cell-type composition of each spot. SPOTlight identified cell-type-specific gene expression signatures from scRNA-seq data, which were subsequently used to deconvolute ST-seq spots. Therefore, the spatiotemporal proportion of all 12 immune cell types identified via scRNA-seq, including monocytes, have been added to the revised Figure 2 and Supplementary Figures 2-5. For example, the monocyte-specific score on day 1 post-MI was as follows:

<Revised supplementary figure 2>

Based on the SPOTlight analysis, the related statements have also been changed as follows:

“We then applied SPOTlight (PMID: 33544846), a nonnegative matrix factorization (NMF)-based spatial deconvolution framework, to infer the cell-type composition of each spot (Fig. 2b). Notably, neutrophils highly infiltrated into the infarcted area on day 1 (41.2% on average) but their numbers were then decreased rapidly (18.7, 10.6, and 9.5% on days 3, 5, and 7, respectively), which was consistent with the scRNA-seq data (Fig. 2b and 2c). On the contrary, macrophages were dispersed across the whole heart rather than clustered in the infarcted area on day 1 (Fig. 2b). However, from day 3, macrophages infiltrated into the infarcted area, and their abundance peaked at late MI (days 5 and 7; 18.8% on average) (Fig. 2c). Monocytes and fibroblasts also infiltrated the infarcted area of the heart in a time-dependent manner

(Supplementary Fig. 1-5). Other immune cell populations, such as B-cell and T-cell populations, were always dispersed across the entire mouse heart (not clustered in the infarcted area), with a very low abundance (Supplementary Fig. 2-5). Together with the scRNA-seq showing immune cell temporal dynamics after MI, ST-seq analysis with deconvolution algorithm further provided their spatial heterogeneity and spatiotemporal dynamics; our ST-seq data clearly showed that monocytes and neutrophils infiltrated into the infarcted area at early MI while macrophages and fibroblasts acted oppositely.” (page 7-8, line 119-135)

Comment 4. In Fig.3B, the authors show a higher presence of resident macrophages at a steady-state, a decrease after MI and restored at days 5-7. Authors need to describe whether the resident macrophages are restored due to proliferation of the remaining resident cells or replacement.

Response: We appreciate this comment. In a recent work by Dick *et al.* (PMID: 30538339), scRNA-seq analysis of mouse heart tissue at steady state identified four distinct macrophage clusters (TIMD4, MHC-II, ISG, and CCR2 M ϕ subsets) as tissue-resident cardiac macrophages. The fate-mapping analysis further showed that CCR2⁻TIMD4⁺ M ϕ were maintained independently of monocytes: CCR2⁻TIMD4⁺MHCII^{hi} M ϕ were partially replaced by monocytes and CCR2⁺MHCII^{hi} M ϕ were fully replaced by monocytes over time. Considering that the differentially expressed genes of TIMD4 M ϕ (*Lyve1*, *F13a1*, *Cbr2*, *Folr2*, and *Timd4*) and MHC-II M ϕ (*H2-Eb1*, *H2-Aa*, *H2-Ab1*) resembled those of clusters 3 (Res-M ϕ 1) and 1 (Res-M ϕ 2), respectively, Res-M ϕ 1 might be restored due to proliferation of the remaining resident cells, while Res-M ϕ 2 might be replaced. To distinguish the monocyte- and embryo-derived resident macrophages more clearly, we have consistently changed the term from tissue-resident macrophages (Res-M ϕ clusters) to steady-state macrophages (SS-M ϕ clusters) in the revised manuscript. These descriptions have been added to the Discussion as follows. “Notably,

SS-Mφ populations decreased after MI and were restored on days 5-7. Considering the previous study, which reported that resident cardiac CCR2⁻TIMD4⁺ Mφ (SS-Mφ1 in this study) were maintained independently of monocytes, while CCR2^{+/+}TIMD4⁻MHCII^{hi} Mφ (SS-Mφ2 in this study) were fully or partially replaced by monocyte over time (PMID: 30538339), SS-Mφ1 might be restored due to proliferation of the remaining resident cells, while SS-Mφ2 might be replaced. Further in-depth analysis using fate-mapping will provide their conclusive origin.” (page 18, line 357-363).

Comment 5. In Fig.5f, expression of CCR2 and Trem2 are inversely correlated, dose CCR2 positive cells are gradually transit to Trem2 macrophages? Or is Trem2 are fresh population at later stage of MI? Moreover, the late macrophages increase at days 5 and 7, don't express CCR but they are not mentioned as resident macrophages. What is their origin?

Response: We agree with the reviewer's concern regarding the origin of Trem2^{hi} macrophages. It is well-known that resident cardiac macrophages are primarily established prenatally and arise from embryonic yolk-sac progenitors. As indicated in the response to comment 4, tissue-resident macrophages are characterized by unique gene signatures such as *Lyve1*, *F13a1*, *Cbr2*, *Folr2*, and *Timd4*. From this point of view, Trem2^{hi} macrophages did not express the tissue-resident macrophage signatures, and their *Ccr2* expression was inversely correlated with Ly6c2^{hi} monocytes. Furthermore, our inferred trajectory of monocytes/macrophages supports a sequential differentiation from Ly6c2^{hi} monocytes to Trem2^{hi} macrophages (revised figure 3c). Therefore, we interpreted that the *Ccr2*-positive cells gradually differentiate into Trem2^{hi} macrophages. To this end, experiments, such as fate-mapping, will be helpful, but this was beyond our capabilities. We have added this limitation to the Discussion as follows: “Further studies are, however, needed to systematically evaluate the Trem2^{hi} macrophage origin, the cells affected by sTrem2, and the related intracellular signaling pathways, to enable a better

understanding of its regenerative effect in infarcted hearts, before conducting human clinical trials” (page 19, line 397-400)

Comment 6. Authors provide IHC image and co-staining of Trem2 and cardiac macrophages, however, whether macrophages are only cell source of Trem2 is not clear. Measurement of Trem2 expression in all cardiac cell fraction is recommended.

Response: As suggested, we performed Trem2 and cardiac macrophage marker (CD68) co-staining in all cardiac cell fractions on day 5 post-MI. The CD68⁺ macrophage proportions increased markedly in the infarcted area and the Trem2 signals overlapped with those of CD68⁺ cells, suggesting that the macrophages are the major cell source of Trem2. We have added these results to the revised manuscript (revised Figure 5c).

<Revised figure 5c>

Comment 7. Please add the quantification of WB in Fig.5b. Additionally, it is required a better

description of the full length and soluble form of Trem2.

Response: As the reviewer suggested, we have added the quantification data for the comparison of relative Trem2 levels and changed “upper bands from Trem2” to “full-length Trem2” in the revised Figure 5b. The description of total Trem2 expression has also been added to the Results section as follows: “The expression of total Trem2 (full-length Trem2 and sTrem2) peaked on day 5 (Fig. 5b).” (page 13, line 269-270).

<Revised figure 5b>

Comment 8. Authors showed soluble fraction of Trem2 increased in later stage of MI. What about in secretome or plasma, is Trem2 commonly detected in animal with MI or heart failure patient?

Response: To answer this, we have compared the sTrem2 levels in the plasma of the MI mouse model (see the figure below). sTrem2 levels increased gradually until 5 days post-MI and then gradually decreased. This trend was largely compatible with that of Trem2-expressing macrophages in the infarcted heart tissue. However, this result did not show statistical significance, so further studies are needed to verify the potential of sTrem2 as a biomarker of MI progression. We did not include this result in the revised manuscript because of the word limitation, and also because this is not the main point of this study.

Comment 9. In Fig.5c, it would be interesting to integrate the staining for cardiomyocytes. Considering the starting demonstrations in which late macrophages co-localize with the damaged area (Fig.3E), a staining to highlight the infarcted area should be performed in combination with CD68 and Trem2.

Response: As the reviewer suggested, we performed Trem2 and cardiomyocyte marker (myoglobin) co-staining in the infarcted heart on day 5 post-MI. However, as shown in the revised Figure 5c, Trem2 expression was mostly observed in the macrophages and there was no significant Trem2 expression in the cardiomyocytes (see the figure below). Therefore, the Trem2 and CD68 co-staining images have been added to the revised Figure 5c to highlight the infarcted area.

Comment 10. Fig.6 should be integrated with additional information such as ratio LV mass/BW and a table with morphometric and echocardiographic characteristics of the mice (mortality rate, infarct size, body weight, IVSd, IVSs, LVIDd, LVIDs; etc.). Table would be great addition.

Response: As the reviewer suggested, we have added the survival rate of each group to the revised Figure 6 and the echocardiography parameters of the mice to the revised Supplementary Table 6. However, we were unable to provide the LV mass/BW ratio because we did not measure LV mass or BW. The related description has also been added to the Results section as follows. “The infarct size was also significantly smaller in sTrem2-GH-treated MI mice (26.0%) than that in PBS- (48.4%) and GH-treated (38.0%) mice (Fig. 6e) and an increased survival rate was observed in the mice treated with sTrem2-GH compared with other groups (Fig. 6f).” (page 15-16, line 323-326)

<Revised figure 6f>

	PBS	GH	sTrem2-GH
IVSd	0.28 ± 0.66	0.29 ± 0.04 **	0.36 ± 0.07 ***
LVIDd	5.97 ± 0.53	5.62 ± 0.67	5.31 ± 0.20 ***
LVPWd	0.53 ± 0.09	0.57 ± 0.03	0.54 ± 0.04
IVSs	0.31 ± 0.01	0.34 ± 0.07	0.43 ± 0.08
LVIDs	5.55 ± 0.58	5.07 ± 0.78	4.21 ± 0.33
LVPWs	0.63 ± 0.02	0.65 ± 0.06	0.65 ± 0.05

<Revised supplementary table 6>

Comment 11. Does sTrem2 with GH specifically targeting macrophages? A distribution of the Trem2 with GH in myocardium should be presented. Moreover, how authors select dose of Trem2 need to be better described.

Response: We have previously verified an injectable gelatin-based hydrogel (GH) as an efficient drug or cell delivery platform for cardiac repair after MI (ACS Appl. Bio Mater. 2020, 3, 1646–1655). Using this method, we have already identified that intramyocardial

transplantation of GH achieves a substantial distribution in the injected region for 7 days, providing a satisfactory microenvironment in infarcted hearts (see the figure below). Accordingly, we have applied GH combined with sTrem2 to peri-infarcted areas based on our optimized protocol. Consistent with the sTrem2 release profiles *in vitro* in the revised Supplementary Figure 13, it may lead to sustained Trem2 engraftment and release due to *in vivo* degradation of GH hydrogels in infarcted areas as described previously. However, despite your valuable comment, we decided not to include similar experiments. In addition, sTrem2 labeling is technically too difficult to include in this revision. For a better understanding of sTrem2 and its regenerative effect, further intensive studies including the distribution of injected sTrem2 will be conducted as future research. This statement has been added to the Discussion of the revised manuscript as follows: “Further studies are, however, needed to systematically evaluate the Trem2^{hi} macrophage origin, the cells affected by sTrem2, and the related intracellular signaling pathways, to enable a better understanding of its regenerative effect in infarcted hearts, before conducting human clinical trials.” (page 19, line 397-400)

Regarding the dose of sTrem2, we were able to set the dose of sTrem2 by changing the concentration according to previous studies, which were relevant for direct *in vivo* administration of recombinant sTrem2 to specific tissues, as well as for its intramyocardial injection into infarcted areas (PMID: 30911003 and 24030424). Although the biological responses depending on sTrem2 concentration should be considered, the results supported the proof-of-concept, indicating that sTrem2 promoted functional and structural improvements of

infarcted hearts *in vivo*. As the reviewer suggested, we have added and re-arranged the relevant statement to the Methods of the revised manuscript as follows: “To evaluate the efficacy of sTREM2, mice were injected with PBS, gel, or gel containing 12 µg sTrem2 into the myocardium at the infarct border zone after the LAD ligation. Each treatment was injected at two sites of the peri-infarcted area (10 µL per site), right after MI, as described previously with slight modifications (PMID: 30911003 and 24030424).” (page 22, line 456-460)

Comment 12. The expression of sTrem2 is increased in late stage (7days) of MI, what about in chronic stage such as 28 days. Authors inject sTrem2 right after MI and it showed protective effect, what is mechanism behind? Is it due to transition of macrophages phenotype or anti-inflammatory response to entire heart? Authors need more evidence to clarify the cardioprotective effect of Trem2.

Response: Thank you for this constructive comment. As the reviewer suggested, we have analyzed the long-term changes of sTrem2 expression in the infarcted heart tissues. Western blotting showed that the expression of sTrem2 was significantly decreased in the chronic stage compared with that in the late stage (see the figure below). It is noteworthy that these initial changes in Trem2 expression observed in the infarcted areas for 7 days are closely related to the inflammation-modulating role and anti-remodeling effect *in vivo*.

Although we found the beneficial anti-remodeling effect of sTrem2 post-MI, it will be a

difficult task to investigate the underlying mechanisms, such as what type of cells are affected by sTrem2 and how sTrem2 induces intracellular signaling. Late-stage Trem2 macrophages may secrete sTrem2 and can affect the macrophage itself via autocrine regulation. To investigate how sTrem2 affects the macrophage, we tested the polarizing effect of sTrem2 on thioglycolate-elicited peritoneal macrophages from C57BL/6 mice. When peritoneal macrophages were activated toward a pro-inflammatory phenotype by lipopolysaccharide (LPS, 1 $\mu\text{g}/\text{mL}$) and interferon- γ (IFN- γ , 4 ng/mL) treatments, *IL-12B* and *iNOS* (pro-inflammatory markers) were upregulated while *Arg1* (anti-inflammatory marker) was downregulated. On the contrary, sTrem2 (200 ng/ml) treatment after LPS and IFN- γ treatments decreased or increased the expression levels of pro-inflammatory and anti-inflammatory markers, respectively (see the figure below). This finding suggests that sTrem2 affects macrophage polarization, but since further detailed research should be performed to draw a firm conclusion, we did not include these *in vitro* data in the revised manuscript. We agree with the reviewer's concern about the Trem2 function in MI, but the exploration of more possibilities is beyond our capacity at this time and will be carried out in future studies. This statement has been added to the Discussion section of the revised manuscript as follows: "Further studies are, however, needed to systematically evaluate the Trem2^{hi} macrophage origin, the cells affected by sTrem2, and the related intracellular signaling pathways, to enable a better understanding of its regenerative effect in infarcted hearts, before conducting human clinical trials." (page 19, line 397-400)

Comment 13. Trem^{hi} macrophages exhibit more M2 phenotypes with increased Arg1 level, however this result is not sufficient to conclude Trem^{hi} macrophages are anti-inflammatory macrophages. Facs analysis of Trem^{hi} macrophages are required for better understanding of the effect of Trem on macrophage polarization.

Response: We appreciate this comment. As the reviewer suggested, we have further analyzed the polarizing status of macrophage subsets by quantifying inflammatory and anti-inflammatory mRNA expression levels in the Ccr2^{hi} and Trem2^{hi} macrophages (revised Fig. 5i and 5j). Ccr2^{hi} macrophages exhibited higher expression levels of pro-inflammatory cytokines (*IL-1 β* and *IL-6*) and chemokines (*Ccl2*, *Cxcr3*, and *Cxcr7*), whereas Trem2^{hi} macrophages expressed higher levels of anti-inflammatory cytokines (*IL-4*, *IL-10*, and *Alox15*), chemokines (*Ccl1* and *Cxcr1*), as well as the anti-inflammatory signature gene, *TGF- β* .

<Revised figure 5i and 5j>

Consistent with our conclusions, these data suggest that Trem2^{hi} macrophages possess anti-inflammatory characteristics. Related statements have been added and to the Results section as follows: “Next, we quantified inflammatory and anti-inflammatory mRNA expression levels in the Ccr2^{hi} and Trem2^{hi} macrophages. Ccr2^{hi} macrophages exhibited higher expression levels of pro-inflammatory cytokines (*IL-1 β* and *IL-6*) and chemokines (*Ccl2*, *Cxcr3*, and *Cxcr7*), whereas Trem2^{hi} macrophages expressed higher levels of anti-inflammatory cytokines (*IL-4*, *IL-10*, and *Alox15*), chemokines (*Ccl1* and *Cxcr1*), as well as the anti-inflammatory signature

gene, *TGF-β* (Fig. 5i and 5j).” (page 14-15, line 296-302) The following were added to the Discussion: “fourth, in the late phase of MI, the number of Trem2^{hi} macrophages abundantly expressing anti-inflammatory signature genes is significantly increased in the infarcted heart tissues and the soluble form of Trem2, the expression of which increased after the peak of full-length Trem2 expression, significantly improves remodeling and cardiac function in the infarcted heart.” (page 17, line 337-341) “Of note, a higher expression of anti-inflammatory signature genes was observed in Trem2^{hi} macrophages, suggesting anti-inflammatory macrophage characteristics.” (page 19, line 383-385)

Comment 14. Despite M2 macrophages present anti-inflammatory characters, they often show also pro-fibrotic features. Authors didn't show any results related to fibrosis or fibroblasts. Additional data about the effect of sTrem2 on fibrosis are required.

Response: As the reviewer suggested, we have further analyzed the expression levels of *TGF-β* and osteopontin (*Spp1*), which are related to the pro-fibrotic potential in regulating post-MI LV remodeling (PMID: 11375279 and 31411676). As shown in the revised Fig. 5j, Trem2^{hi} macrophages exhibited higher expression levels of *TGF-β* and *Spp1* and the related statements have been added and to the Results section as follows: “Of note, Trem2^{hi} macrophages also exhibited higher expression level of osteopontin (*Spp1*), which is related to pro-fibrotic potential in regulating post-MI LV remodeling.” (page 15, line 302-304)

Comment 15. About Fig.6C, it would be interesting to show the results compared to the control group of mice without LAD ligation. Moreover, it seems that the gelatin hydrogel without Trem2 significantly improves the LVEF and FS compared to the PBS treatment. Authors need clarify that in discussion.

Response: As the reviewer suggested, we have added the information on the control group to

the revised Figure 6c and 6d.

<Revised figure 6c and 6d>

Regarding the cardiac function improvements in the GH-treated group without sTrem2, we have previously verified the effects of this gelatin-based hydrogel (GH) as already mentioned in the response to comment 11 (ACS Appl. Bio Mater. 2020, 3, 1646–1655). In our previous work, GH reduced the infarct size compared with the PBS-treated control, suggesting the potential impact of physical support to prevent infarct extension and LV dilation, which is consistent with the current results. Thus, GH hydrogels not only provide physical support for the infarcted myocardium but are also a suitable microenvironment for sTrem2 engraftment and its sustained release contributing to the cardiac regenerative effect. This statement has been added to the Discussion of the revised manuscript as follows: “Importantly, when we used an injectable gelatin-based hydrogel, which provided physical support for the infarcted myocardium and an appropriate microenvironment for simultaneously delivering biomolecules (ACS Appl. Bio Mater. 2020, 3, 1646–1655), the injection of sTrem2 with GH into the peri-infarct area of MI in mice resulted in higher LVEF/FS, lower ESV, and morphologically less dilated and well-remodeled LV than those in control animals, supporting the inflammation-modulating role and anti-remodeling effect of Trem2 *in vivo*.” (page 19, line 391-397)

Comment 16. (Minor points) The authors should describe the number of biological replicates for each experiment.

Response: As the reviewer suggested, we have added the number of biological replicates for each experiment to Figure legend.

Comment 17. (Minor points) In the introduction is mentioned the nomenclature M1 and M2 that is obsolete. The *in vivo* environment is more complex, as demonstrated by this paper, so this nomenclature can be limited only to an *in vitro* environment. I would suggest to the authors to specify this or to edit it (PMID: 27458196).

Response: We appreciate this comment. As suggested, the M1/M2 dichotomy is no longer used in the revised manuscript and related sentences were consistently changed in the Introduction as follows: “However, this dichotomous classification was obsolete, as *in vivo* environment of macrophages are more complex (PMID: 27458196). Therefore, a more precise evaluation is needed for accurate characterization of the dynamics of macrophage heterogeneity during the acute period of MI.” (page 4, line 61-64)

The following was added to the Discussion: “third, the single-cell trajectory analysis supports a sequential differentiation from Ly6c^{hi} monocytes to Late-Mφ rather than the obsolete dichotomous M1-M2 paradigm.” (page 17, line 335-337) Filip K. Swirski, the author of the paper the reviewer mentioned and one of the authors of this manuscript, has read and approved this manuscript.

Comment 18. (Minor points) Since Trem2 is the crucial molecule discussed in this manuscript, this should be mentioned and briefly described also in the introductory part, not only in the discussions.

Response: As the reviewer suggested, we have added the brief description to the Introduction

as follows: “We identified a novel macrophage subset, Trem2^{hi} macrophages, with anti-inflammatory characteristics, specifically dominant in late MI in the infarcted heart. Moreover, *in vivo* injection of soluble Trem2 led to significant functional and structural improvements in infarcted hearts.” (page 5, line 70-73)

Comment 19. (Minor points) The results of Fig.1 and Fig.2 should be better discussed. For Fig.2 the conclusion is completely missed.

Response: As the reviewer suggested, we have added the conclusions to the Results section as follows: “Together with the scRNA-seq showing immune cell temporal dynamics after MI, ST-seq analysis with deconvolution algorithm further provided their spatial heterogeneity and spatiotemporal dynamics; our ST-seq data clearly showed that monocytes and neutrophils infiltrated into the infarcted area at early MI while macrophages and fibroblasts acted oppositely.” (page 8, line 131-135)

Comment 20. (Minor points) Page 11, line 244, there is a mistake in the mentioned Figure. It should be 5c and not 5b.

Response: We have corrected this mistake in the revised manuscript.

Comment 21. (Minor points) Fig.5d-f requires an extensive explanation.

Response: As the reviewer suggested, we have added the additional explanation to the Results section of the revised manuscript as follows: “The proportion of Ccr2^{hi} macrophages rapidly increased at day 1 post-MI and then declined over time (Fig. 5d), suggesting that monocyte-derived macrophages were recruited to the infarcted areas at early MI, which was consistent with a sub-clustering analysis of scRNA-seq data. In contrast, the proportion of Trem2^{hi} macrophages began to increase at day 3 and peaked 5 days after MI, indicating a specific

dominance in late MI in the infarcted heart (Fig. 5e-f).” (page 14, line 286-291)

Comment 22. (Minor points) In material and methods, paragraph co-localization assay, the reference of the secondary antibodies is missing.

Response: We have added the secondary antibody information (page 27, line 587-588).

REVIEWER COMMENTS

Reviewer #1 (Remarks to the Author):

With additional analyses, the manuscript has been improved somewhat. However, it still remains unclear whether Trem2+ macrophages are anti-inflammatory and sTrem2 is a key mediator produced by macrophages that control remodeling processes after MI. Additional analytical data now show that Trem2 is expressed in subsets of macrophages that express higher levels of Il10, Tgfb1, and Il4. Though authors note that they are anti-inflammatory macrophages, the expression profile suggests that they are also pro-fibrotic and pro-remodeling. Though ST-seq data are of interest, the mechanistic side of the study is very limited and provides insufficient evidence for the beneficial role of Trem2. Authors now abandoned M1/M2 dichotomy. However, they still rely on simple labeling of anti-inflammatory due to the expression of some signature genes, such as Arg1.

p. 14-15 "Ccr2hi macrophages exhibited higher expression levels of pro-inflammatory cytokines (IL-1 β and IL-6) and chemokines (Ccl2, Cxcr3, and Cxcr7), whereas Trem2hi macrophages expressed higher levels of anti-inflammatory cytokines (IL-4, IL-10, and Alox15), chemokines (Ccl1 and Cxcr1), as well as the anti-inflammatory signature gene, TGF- β (Fig. 5i and 5j)."

IL-4 is not an anti-inflammatory cytokine, and neither is CCL1. Cxcr1 is a receptor, and Alox15 is an enzyme. Please carefully revise these and other misleading and erroneous labeling and explanations of genes and cells.

Expression of the genes suggests that Trem2-expressing macrophages contribute to ECM production/fibrosis in MI. This appears to be contradictory to the authors' claim that Trem2 ameliorates structural remodeling after MI. Previous studies have demonstrated that macrophages crucially contribute to scar formation. Though during scar formation and remodeling, they may also regulate inflammation, their potential function to lead remodeling should not be ignored.

Minor:

Gene names in graphs should be official gene symbols rather than conventional names (e.g., Fig. 5i, j).

Reviewer #2 (Remarks to the Author):

Jung and colleagues have revised some of their statements that largely addresses my prior questions and concerns. The major issue was that ST-seq is not technically single-cell and this caveat needed to be discussed in the manuscript, which the authors have now done. The authors have also reanalyzed their data using SPOTlight in order to better infer the cell-type composition of each spot. While it does not fully address the caveats (and frankly the non-single-cell nature of the approach cannot be solved), it is a reasonable effort on their part. I would also note that the authors have performed new experiments and analyses to address the concerns. Thus, the clarifications and new data that they have provided have improved the manuscript, which I believe would be of interest to the field and broad readership of Nature Communications.

Reviewer #3 (Remarks to the Author):

Single-cell transcriptomics unveils the spatiotemporal dynamics of macrophage heterogeneity and a potential role of Trem2hi macrophages in infarcted hearts

The authors addressed my previous concerns nicely. However, I have a few additional comments.

1. Although authors showed nicely the anti-inflammatory phenotype of Trem2hi macrophages, still the connection between Trem2hi and soluble form of Trem2 (sTrem2) is not clear. Please describe more detail how you come to study soluble form of Trem2 in the result section. For better understanding, the in vitro data of treatment with sTrem2 in peritoneal macrophages (showed in

rebuttal letter) should be added in supplementary figures.

2. Regarding origins of sTrem2, authors claimed that "Late--stage Trem2 macrophages may secrete sTrem2 and can affect the macrophage itself via autocrine regulation." This can be simply checked by secreted level of sTrem2 from Trem2hi macrophages or Trem2 overexpression macrophages in vitro.

3. Post-MI cardiac function is directly influenced by fibrosis, the indicator of stiffness of heart. While the effect of sTrem2 on improved cardiac function is well described, the contribution into global fibrosis is totally lack in this manuscript. Especially, reparative macrophages have been often shown to participate in fibrosis development, therefore, in addition to cardiac function, fibrosis measurement has to be added.

4. Authors' in vivo data looks really promising, especially there's big difference in survival rate between PBS and Trem2-GH. But sTrem2 is already high in d7 post-MI, is it not sufficient or not early enough? How's sTrem2 level in heart at d7 post-MI upon sTrem2 injection? In line with this, could you also provide PK study of sTrem2?

5. Minor, page 15, line 303, correct typo "osteropontin".

Response to reviewer comments

Reviewer #1

Comment 1. p. 14-15 “Ccr2^{hi} macrophages exhibited higher expression levels of pro-inflammatory cytokines (IL-1 β and IL-6) and chemokines (*Ccl2*, *Cxcr3*, and *Cxcr7*), whereas Trem2^{hi} macrophages expressed higher levels of anti-inflammatory cytokines (IL-4, IL-10, and *Alox15*), chemokines (*Ccl1* and *Cxcr1*), as well as the anti-inflammatory signature gene, TGF- β (Fig. 5i and 5j).” IL-4 is not an anti-inflammatory cytokine, and neither is CCL1. *Cxcr1* is a receptor, and *Alox15* is an enzyme. Please carefully revise these and other misleading and erroneous labeling and explanations of genes and cells.

Response: As suggested, we have removed the *Il4* and *Ccl1*, and carefully corrected the misclassifications as follows. “In the quantitative reverse transcription-PCR analysis, Ccr2^{hi} macrophages exhibited higher expression levels of pro-inflammatory genes (cytokines: *Il1b* and *Il6*, chemokine: *Ccl2*, and chemokine receptors: *Cxcr3* and *Cxcr7*), whereas Trem2^{hi} macrophages expressed higher levels of anti-inflammatory genes (cytokines: *Il10* and *Tgfb1*, chemokine receptor: *Cx3cr1*, and enzyme: *Alox15*) (Fig. 5i and 5j).” (page 14, line 293-298)

Comment 2. Expression of the genes suggests that Trem2-expressing macrophages contribute to ECM production/fibrosis in MI. This appears to be contradictory to the authors’ claim that Trem2 ameliorates structural remodeling after MI. Previous studies have demonstrated that macrophages crucially contribute to scar formation. Though during scar formation and remodeling, they may also regulate inflammation, their potential function to lead remodeling should not be ignored.

Response: We appreciate the reviewer’s valuable comment. It is well-known that cardiac repair after MI results from a finely orchestrated and complex series of events, initiated by an inflammatory phase (\approx 3–4 day in mice) with intense inflammatory responses, followed by a

reparative phase (peak day ≈ 7 post-MI) with a resolution of inflammation, myofibroblast proliferation, and scar formation over the next several days, presenting increased expression of anti-inflammatory and profibrotic factors (eg, *Il10* and *Tgfb1*). Therefore, the higher expression level of *Il10* and *Tgfb1* on days 5 and 7 post-MI seems reasonable. In addition, because it is not clear whether the increase in fibrosis-related gene expression leads to actual fibrosis development, we have further analyzed the wall thickness and collagen area in the infarcted heart tissues on day 28 post-MI. The MI mice treated with sTrem2 exhibited significantly thicker left ventricular wall and reduced collagen area in the infarcted region than the mice treated with PBS or GH alone, indicating that a local injection of sTrem2 induced a favorable LV remodeling after MI. The related statement has been added to the Result as follow: “Histologically, the mice treated with sTrem2-GH showed less dilated and well-remodeled LV with thicker infarcted walls and lower fibrosis (Fig. 6b and Supplementary Fig. 16).” (page 15, line 320-322)

<Supplementary figure 16; n=4-7, * $P < 0.05$, ns: not significant>

Comment 3. (Minor points) Gene names in graphs should be official gene symbols rather than conventional names (e.g., Fig. 5i, j).

Response: As suggested, gene names have been consistently changed to official gene symbols in the revised manuscript. Thank you.

Reviewer #3

Comment 1. Although authors showed nicely the anti-inflammatory phenotype of Trem2^{hi} macrophages, still the connection between Trem2^{hi} and soluble form of Trem2 (sTrem2) is not clear. Please describe more detail how you come to study soluble form of Trem2 in the result section. For better understanding, the *in vitro* data of treatment with sTrem2 in peritoneal macrophages (showed in rebuttal letter) should be added in supplementary figures.

Response: As the reviewer suggested, we have added the *in vitro* data as a supplementary figure with relevant statements in the Result section as follows: “We further investigated the sTrem2 effects on macrophage polarization using thioglycolate-elicited peritoneal macrophages. sTrem2 treated pro-inflammatory macrophages showed decreased *Il12b* and *Nos2* expression (pro-inflammatory markers) and increased *Arg1* expression, suggesting that sTrem2 affects macrophage polarization (Supplementary Fig. 14).” (page 15, line 301-304)

Detailed methods are described in the supplementary figure legend as follows: “Thioglycolate-elicited peritoneal macrophages from C57BL/6 mice were sorted and polarized to a pro-inflammatory phenotype by lipopolysaccharide (LPS, 1 μg/mL) and interferon-γ (IFN-γ, 4 ng/mL) treatments. The expression levels of *Il12b* and *Nos2* (pro-inflammatory markers) were upregulated while *Arg1* (anti-inflammatory marker) was downregulated in a pro-inflammatory phenotype. On the contrary, sTrem2 (200 ng/mL) treated pro-inflammatory phenotype macrophages showed decreased expression levels of *Il12b* and *Nos2* and increased expression levels of *Arg1*.”

<Supplementary figure 14; n=4, *P<0.05, **P<0.01, and ***P<0.001>

Comment 2. Regarding origins of sTrem2, authors claimed that “Late--stage Trem2 macrophages may secrete sTrem2 and can affect the macrophage itself via autocrine regulation.” This can be simply checked by secreted level of sTrem2 from Trem2^{hi} macrophages or Trem2 overexpression macrophages *in vitro*.

Response: As the reviewer suggested, we have evaluated the secretion levels of sTrem2 in Trem2^{hi} and Trem2^{low} macrophages in this revision. For this, F4/80⁺Trem2^{hi} and F4/80⁺Trem2^{low} macrophage subsets were sorted from the infarcted heart tissue on days 3 (Trem2^{hi} macrophages began to increase) and 5 (Trem2^{hi} macrophages reached a peak) post-MI. The sorted cells were cultured for 2 hours and the secretion levels of sTrem2 was measured by ELISA. When we compared the secretion levels of sTrem2 between Trem2^{hi} and Trem2^{low} macrophages, sTrem2 was detected only in Trem2^{hi} macrophages (see below figure). Of note, the secretion level of sTrem2 on day 5 was significantly higher than that on day 3, which was consistent with western blotting analysis. This data has been added to a supplementary figure with relevant statements in the Result section as follows: “We also evaluated the secretion levels of sTrem2 in Trem2^{hi} and Trem2^{low} macrophages from days 3 and 5 post-MI and identified that sTrem2 was secreted only in Trem2^{hi} macrophages (Supplementary Fig. 12). Of note, the secretion level of sTrem2 on day 5 was significantly higher than that on day 3. Collectively, these results indicate that Trem2 is highly expressed by macrophages in the infarcted heart and sTrem2 is exclusively secreted from Trem2^{hi} macrophages.” (page 13-14, line 271-276)

<Supplementary figure 12; n=6-8, ** P <0.01, n.d.: not detected>

Comment 3. Post-MI cardiac function is directly influenced by fibrosis, the indicator of stiffness of heart. While the effect of sTrem2 on improved cardiac function is well described, the contribution into global fibrosis is totally lack in this manuscript. Especially, reparative macrophages have been often shown to participate in fibrosis development, therefore, in addition to cardiac function, fibrosis measurement has to be added.

Response: We appreciate this valuable comment. To address this issue, we have further analyzed the wall thickness and collagen area in the infarcted heart tissues on day 28 post-MI. The MI mice treated with sTrem2 exhibited significantly thicker left ventricular wall and reduced collagen area in the infarcted region than the mice treated with PBS or GH alone,

indicating that a local injection of sTrem2 induced a favorable LV remodeling after MI. The related statement has been added to the Result as follow: “Histologically, the mice treated with sTrem2-GH showed less dilated and well-remodeled LV with thicker infarcted walls and lower fibrosis (Fig. 6b and Supplementary Fig. 16).” (page 15, line 320-322)

<Supplementary figure 16; n=4-7, * P <0.05, ns: not significant>

Comment 4. Authors' *in vivo* data looks really promising, especially there's big difference in survival rate between PBS and Trem2-GH. But sTrem2 is already high in d7 post-MI, is it not sufficient or not early enough? How's sTrem2 level in heart at d7 post-MI upon sTrem2 injection? In line with this, could you also provide PK study of sTrem2?

Response: We agree with the reviewer's concern about the PK of sTrem2. To this end, a multiple tracking system for injected sTrem2 and secreted sTrem2 by macrophages should be

developed, which is however beyond our capabilities and the scope of this study at this time.
This limitation already described in the Discussion section (page 19, line 396-399).

Comment 5. (Minor points) page 15, line 303, correct typo “osteropontin”.

Response: We have corrected the typo in this revision. Thank you.

REVIEWERS' COMMENTS

Reviewer #1 (Remarks to the Author):

The authors have resolved the issues I raised.

Reviewer #3 (Remarks to the Author):

The authors nicely addressed my remaining points.